

**Leveraging 35 years of forest research in the southeastern U.S. to constrain carbon**
**cycle predictions: regional data assimilation using ecosystem experiments**
R. Quinn Thomas[1*], Evan Brooks[1], Annika Jersild[1], Eric Ward[2], Randolph Wynne[1], Timothy J.
Albaugh[1], Heather Dinon Aldridge[3], Harold E. Burkhart[1], Jean-Christophe Domec[4,5], Thomas R.
Fox[1], Carlos A. Gonzalez-Benecke[6], Timothy A. Martin[7], Asko Noormets[8], David A. Sampson[9],
Robert O. Teskey[10]
[1]Department of Forest Resources and Environmental Conservation, Virginia Tech, USA,
[2]Climate Change Science Institute and Environmental Sciences Division, Oak Ridge National
Laboratory, USA,
[3]State Climate Office of North Carolina, North Carolina State University, USA,
[4]Bordeaux Sciences Agro, UMR 1391 INRA-ISPA, 33175 Gradignan Cedex, France,
[5]Nicholas School of the Environment, Box 90328, Duke University, Durham, NC 27708, USA,
[6]Department of Forest Engineering, Resources and Management, Oregon State University, USA,
[7]School of Forest Resources and Conservation, University of Florida, USA,
[8]Department of Forestry and Environmental Resources, North Carolina State University, USA,
[9]Decision Center for a Desert City, Arizona State University, USA,
[10]Warnell School of Forestry and Natural Resources, University of Georgia, USA,
*Corresponding author: R. Quinn Thomas (rqthomas@vt.edu)



**Abstract**
Predicting how forest carbon cycling will change in response to climate change and management
depends on the collective knowledge from measurements across environmental gradients,
ecosystem manipulations of global change factors, and mathematical models. Formally
integrating these sources of knowledge through data assimilation, or model-data fusion, allows
the use of past observations to constrain model parameters and estimate prediction uncertainty.
However, the influence of different experimental treatments on those predictions depends on the
exact methods and techniques used for data assimilation. Here, we introduce a hierarchical
Bayesian DA approach (Data Assimilation of Pine Plantation Ecosystem Research, DAPPER)
that uses observations of carbon stocks, carbon fluxes, water fluxes, and vegetation dynamics
from loblolly pine plantation ecosystems across the Southeastern U.S. to constrain parameters in
a modified version of the 3-PG forest growth model. The observations included major
experiments that manipulated atmospheric carbon dioxide ($CO_2$) concentration, water, and
nutrients, along with non-experimental studies that spanned environmental gradients across an
$8.6 \times 10^5$ km$^2$ region. We optimized regionally representative posterior distributions for the most
sensitive model parameters, which dependably predicted data from plots withheld from the data
assimilation. The posterior distributions of parameters associated with ecosystem responses to
$CO_2$, precipitation, and nutrient addition, along with the corresponding regional changes in
production associated with nutrient fertilization and drought, depended on how the experimental
data were assimilated. In particular, assimilating nutrient addition experiments reduced the
predicted sensitivity to nutrient fertilization while assimilated water manipulation experiments
increased the sensitivity to drought. Further, it was necessary to assimilate data from the $CO_2$
experimental enrichment site before other studies to constrain the parameters associated with the



influence of $CO_2$ on canopy photosynthesis. The ambient $CO_2$ plots were numerous and had a
large contribution to the cost function compared to the low number of elevated $CO_2$ plots (289
ambient vs. 5 elevated plots). Overall, we demonstrated how three decades of research in
southeastern U.S. planted pine forests can be used to develop data assimilation techniques that
use multiple locations, multiple data streams, and multiple ecosystem experiment types to
optimize parameters. This approach allows for future predictions to be consistent with a rich
history of ecosystem research across a region.



## 1 Introduction

Forest ecosystems absorb and store a large fraction of anthropogenic carbon dioxide ($CO_2$)

emissions (Le Quere et al., 2015; Pan et al., 2011) and supply wood products to a growing

human population (Shvidenko et al., 2005). Therefore, predicting future carbon sequestration and

timber supply is critical for adapting forest management practices to future environmental

conditions and for using forests to assist with reduction of atmospheric $CO_2$ concentrations. The

key sources of information for developing these predictions are results from global change

ecosystem manipulation experiments, observations of forest dynamics across environmental

gradients, and process-based ecosystem models. The challenge is integrating these three tools

into a common framework for creating probabilistic predictions, or forecasts (based on (Luo et

al., 2011a)), that provide information on both the expected future state of the forest and the

probability distribution of those future states.

Data assimilation (DA), or data-model fusion, is an increasingly used framework for integrating

ecosystem observations into ecosystem models (Luo et al., 2011a; Niu et al., 2014; Williams et

al., 2005). DA integrates observations with ecosystem models through statistical, often Bayesian,

methods that generate probability distributions for ecosystem model parameters and initial states.

DA allows for the explicit accounting of observational uncertainty (Keenan et al., 2011), the

incorporation of multiple types of observations with different time scales of collection

(Richardson et al., 2010), and the representation of prior knowledge through informed parameter

prior distributions or specific relationships among parameters (Bloom and Williams, 2015).

Using DA to parameterize ecosystem models with observations from multiple locations that

leverage environmental gradients and from ecosystem manipulation experiments will allow for



forecasts to be consistent with the rich history of global change research in forest ecosystems.

Ecosystem manipulation experiments provide a controlled environment in which data collected
can be used to describe how forests acclimate and operate under altered environmental
conditions (Medlyn et al., 2015). These data may be used to constrain model parameters that are
associated with specific physiological functions associated with, for example, carbon allocation
and turnover as related to the controlled manipulation. Furthermore, the assimilation of
experiments may increase parameter identifiability (reducing equifinality (Luo et al., 2009)),
where two parameters have compensating controls on the same processes, by isolating the
response to a manipulated driver. For example, carbon assimilation and primary productivity can
be modeled as a light and temperature controlled process that is adjusted by nutrients, water, and
atmospheric $CO_2$ concentration. In this case, the productivity may mathematically be equal
between a parameterization that has high potential conversion of light to photosynthesis (high
quantum yield) but low relative nutrient availability and a parameterization with low quantum
yield but high relative nutrient availability. Therefore, the challenge is that the same rate of
production can emerge from different contributions of environmental controls.

For future predictions with changing environmental conditions, the relative contribution of each
environmental control should be separated in order to correctly parameterize the sensitivity to
changes in the environment. Key examples of existing and past ecosystem experiments that have
the potential to isolate specific parameters in DA include $CO_2$ enrichment, water manipulation,
nutrient addition, and elevated soil temperature experiments. Many of these experiments are
common, particularly when including nutrient addition experiments in managed forests. Other



types of experiments are less common, but the few sites with the experiments, such as whole-
ecosystem $CO_2$ enrichment, include intensive measurements of numerous carbon pools and
fluxes required for model optimization.


Developing optimized parameters that apply to a region requires assimilating observations that
span environmental gradients to support the application of model predictions to a range of
climatic conditions, nutrient availabilities, and soil water dynamics. Therefore, the DA of
multiple research sites across a region is an important extension of prior DA research focused on
DA at a single site with multiple types of observations (Keenan et al., 2012; Richardson et al.,
2010; Weng and Luo, 2011). Incorporating multiple locations that include global change
experiments in DA is associated with numerous challenges. First, prior research has
demonstrated that high frequency observations (i.e., daily, or more frequent, net ecosystem
exchange observations) can overwhelm the contribution of low frequency observations (i.e.,
annual tree diameter measurements) to the cost-function used for optimization (Richardson et al.,
2010), resulting in a parameter set that predominately represents the high-frequency dynamics.
DA of ecosystem experiments and regional observations can present similar issues because key
contrasts isolated in an ecosystem experiment with relatively few plots may be overwhelmed by
the contribution of more numerous regional observations from non-manipulated plots. For
example, whole ecosystem $CO_2$ enrichment experiments are uncommon but are the only
observations representing ecosystem dynamics in an environment with over 550 ppm
atmospheric $CO_2$ (McCarthy et al., 2010). Therefore, DA techniques may be required that assign
additional weight to unique, but rare, experiments in the DA approach. As an example, a multi-



stage Bayesian approach could be used where the observations from the unique experiment are
assimilated first and the posteriors from that assimilation are used as priors for the assimilation
of the remaining observations. Second, DA requires using highly simplified ecosystem models
because many DA methods use millions of iterations to explore parameter distributions and these
iterations have to be applied to both control and manipulated treatments. However, in tension
with the need for simple models in DA, more complex models that simulate carbon, water, and
nutrient dynamics are also needed to fully leverage the diversity of ecosystem manipulation
experiments. Monthly time-scale models of ecosystem processes may be well suited to overcome
these challenges for application to predicting changes in biomass over decades in response to
global change. First, the contribution of monthly flux and annual biomass measurements to the
optimized cost function is more similar in monthly than daily models (12:1 vs. 365:1). Second,
they are computationally more efficient than daily models commonly used in DA, allowing data
spanning hundreds of plots and multiple decades to be assimilated. Finally, DA is able to
calibrate parameters associated with carbon, nitrogen, and water cycles so that they are
appropriate for an aggregated monthly time step, helping prevent potential issues associated
when applying daily parameterizations to coarser temporal time-steps.

Southeastern U.S. planted pine forests are ideal ecosystems for exploring the application of DA
to carbon cycle and forest production predictions. These ecosystems are dominated by loblolly
pine (*Pinus taeda* L.), thus allowing for a single parameter set to be applicable to a large region
containing many soil types and climatic gradients. Loblolly pine represents more than one half of
the standing pine volume in the southern United States (11.7 million ha) and is by far the single
most commercially important forest tree species for the region, with more than 1 billion





seedlings planted annually (Fox et al., 2007; McKeand et al., 2003). There is also a rich history
of experimental research focused on global change factors including region-wide nutrient
addition (Albaugh et al., 2016; Carlson et al., 2014; Raymond et al., 2016), water exclusion
(Bartkowiak et al., 2015; Tang et al., 2004; Ward et al., 2015; Will et al., 2015), and water
addition experiments (Albaugh et al., 2004; Allen et al., 2005; Samuelson et al., 2008). The
region also includes a long-term ecosystem $CO_2$ enrichment study (McCarthy et al., 2010).
Furthermore, many of these experiments are multi-factor with water exclusion-by-nutrients (Will
et al., 2015), water addition-by-nutrients (Albaugh et al., 2004; Allen et al., 2005; Samuelson et
al., 2008), and $CO_2$-by-nutrients treatments (McCarthy et al., 2010; Oren et al., 2001). Beyond
experimental treatments, Southeastern U.S. loblolly pine ecosystems include at least two eddy-
covariance sites with high frequency measurements of carbon and water fluxes along with
biometric observations over many years (Noormets et al., 2010; Novick et al., 2015), and sites
with multi-year sap flow data (Ewers et al., 2001; Gonzalez-Benecke and Martin, 2010; Phillips
and Oren, 2001). Finally, there are available studies that include plots that span the regional
environmental gradients and extend back to the 1980s (Burkhart et al., 1985). Overall, the high
availability of observations of biomass stocks, leaf area index (LAI), carbon fluxes, water fluxes,
and vegetation dynamics that span the past 35 years in loblolly pine ecosystems, including plots
with experimental manipulation and plots across environmental gradients, is well suited to
potentially constrain model parameters and predictions of how carbon cycling responds to
environmental change.

Our objective was to develop a DA approach that integrated diverse data from multiple locations,
including ecosystem experiments, for predicting how forest productivity may respond to global





change. We applied DA techniques to optimize a monthly-time step, simple forest productivity
model using southeastern U.S.-wide experimental (nutrient addition, $CO_2$ enrichment, and water
manipulations) and non-experimental data from 35 years of loblolly pine plantation research in
the region. Our DA approach, DAPPER (Data Assimilation of Pine Plantation Ecosystem
Research), is unique in its focus on simultaneously assimilating observations from multiple
locations, experimental types, and data streams into a simple ecosystem model that includes
carbon, water, and (implicitly) nutrients using a hierarchal Bayesian technique to develop
parameter distributions. We used the DAPPER system to evaluate the sensitivity of biomass
predictions and parameter distributions to the inclusion of ecosystem experiments in DA and to
predict the regional sensitivity of forest production to nutrient fertilization and drought.

**2 Methods**

2.1 Ecosystem Model
We used a modified version of the Physiological Principles Predicting Growth (3-PG) Model to
simulate vegetation dynamics in loblolly pine stands (Bryars et al., 2013; Gonzalez-Benecke et
al., 2016; Landsberg and Waring, 1997). 3-PG is a stand-level vegetation model that runs at the
monthly time-step and includes vegetation carbon dynamics and a simple soil water bucket
model (Figure 1). While a complete description of the 3-PG model and our modifications can be
found in the Supplemental Material, the key concept for interpreting the results is that gross
primary productivity (GPP) was simulated using a light-use efficiency approach where the
absorbed photosynthetically active radiation (APAR) was converted to carbon based on a
quantum yield. Quantum yield was simulated using a parameterized maximum quantum yield



($\alpha$) that was modified by environmental conditions including air temperature, atmospheric $CO_2$,
available soil water and soil fertility. The available soil water and soil fertility modifiers were
values between 0 and 1, while the atmospheric $CO_2$ modifier had a value of 1 at 350 ppm and
values greater than 1 at higher $CO_2$ concentrations.

Elevated $CO_2$ modified tree physiology by increasing quantum yield, based on an increasing but
saturating relationship with atmospheric $CO_2$. We also added a function where the allocation to
foliage relative to stem biomass decreased as atmospheric $CO_2$ increased. Available soil water
and quantum yield were positively related through a logistic relationship between relative
available soil water and the quantum yield modifier, where relative available soil water was the
ratio of simulated available soil water to a plot-level maximum available soil water. Soil fertility
and quantum yield were proportionally related, where quantum yield was scaled by an estimate
of relative stand-level fertility where a value of 1 was the maximum fertility. The fertility
modifier (FR) was constant throughout a simulation of a plot and was either based on site
characteristics or directly optimized as a stand-level parameter. Here we used site-index, a
measure of the height of a stand at a specified age (25 years), and the 35-year mean annual
temperature as site characteristics to predict FR. For a given climate, site index captures
differences in soil fertility, where a lower site index corresponded to a site with lower fertility.
However, regional variation in site index also included the influence of climate on growth rates
that were already accounted for in the other environmental modifiers in the 3-PG model. To
account for the climatic influence on site index, a long-term climate variable (35-year mean
annual temperature) was included in the empirical relationship that predicted FR as an
increasing, but saturating, function of site index. For plots with nutrient fertilization, FR was a





directly optimized parameter. For our application of the 3-PG model using DA, we removed the
previously simulated dependence of total root allocation on FR(Bryars et al., 2013; Gonzalez-
Benecke et al., 2016). Therefore, plots with lower FR could be interpreted to have lower
quantum yield. Other environmental conditions influenced GPP, including temperature, frosts
days, and vapor pressure deficit with a description of these modifiers found in the Supplemental
Material.

Each month, net primary production (a parameterized and constant proportion of GPP) was
allocated to foliage, stem (stemwood, stembark, and branches), coarse roots, and fine roots.
Differing from previous applications of 3-PG to loblolly pine ecosystems, we modified the
model to simulate fine roots and coarse roots separately. 3-PG also simulated simple population
dynamics by including stem density as a state variable. Stem density and stem biomass pools
were reduced by both density-dependent and density-independent mortality (a new
modification), with the former based on the concept of self-thinning. Finally, we added a simple
model of hardwood understory vegetation to enable the use of estimates of gross primary
productivity and evapotranspiration from eddy-covariance tower studies with significant
understories. Details of the model can be found in the Supplemental Material.

The water cycle was a simple bucket model with transpiration predicted using a Penman-
Monteith approach (Bryars et al., 2013; Gonzalez-Benecke et al., 2016; Landsberg and Waring,
1997). The canopy conductance used in the Penman-Monteith subroutine was modified by
environmental conditions. The modifiers include the same available soil water and vapor
pressure deficit modifier as used in the GPP calculation. Maximum canopy conductance



occurred when simulated LAI exceeded a parameterized value of leaf area index (LAI).
Evaporation was equal to the precipitation intercepted by the canopy. Runoff occurred when the
available soil water exceeded a plot-specific maximum available soil water. As in prior
applications of 3-PG, available soil water was not allowed take a value below a minimum
available soil water, resulting in an implicit irrigation in very dry conditions.

The 3-PG model used in this study simulated the monthly change in eleven state variables per
plot: four stocks for loblolly pines, five stocks for understory hardwoods, loblolly pine stem
density (stems ha$^{-1}$), and available soil water. The key fluxes that were used for DA included
monthly GPP, monthly evapotranspiration (ET), annual root turnover, and annual foliage
turnover. In total, 46 parameters were required by 3-PG with 31 of the parameters optimized
using DA (Table 1, Supplemental Table 1, SI Table 2). The model required mean daily
maximum temperature, mean daily minimum temperature, daily PAR, total frost days, total rain
at the monthly time scale, monthly atmospheric $CO_2$, and latitude. Each plot also required
maximum available soil water, site index, mean annual temperature, and the initial condition of
the eleven state variables as model inputs (Figure 2).

**2.2 Observations**
We used thirteen different data streams from 294 plots at 187 unique locations spread across the
region to constrain model parameters (Table 2; Figure 3). The data streams covered the period
between 1981 to 2015. All data streams were not available in all plots (Table 2; Table 3). The
most common set of data streams were annual or less frequent observations of stand stem
biomass (defined as the sum of stemwood, stembark and branches), winter foliage biomass, and
living tree counts. The stem and foliage biomass were optimized using regional allometric



models based on measurements of tree diameter, height, and plot level-stem size distributions
(Gonzalez-Benecke et al., 2014). The most comprehensive set of data streams was from Duke
Forest where annual measurements and allometric-based estimates were made of stem biomass
(loblolly pine and hardwood), coarse root biomass (loblolly pine and hardwood), fine root
biomass (combined loblolly pine and hardwood), stem count (loblolly pine only), leaf turnover
(combined loblolly pine and hardwood), and fine root production (combined loblolly pine and
hardwood). The Duke Forest dataset (DK3 combined with the Duke FACE $CO_2$ fertilization
study) also included monthly observations of LAI, gross ecosystem production (GEP; modeled
gross primary productivity from net ecosystem exchange measured at an eddy-covariance tower),
and ET. The set of data streams associated with a particular site and experimental design is
shown in Table 3. The measurement uncertainty associated with each data stream is listed in
Table 2. Since the model used a monthly time-step, and plots with only biomass and stem density
observations were more common than plots with monthly flux estimates, the data used in the
optimization cost function were not dominated by high frequency data streams (GEP and ET).

**2.3 Data assimilation method**
We used a hierarchal Bayesian framework to approximate the posterior probability distributions
of model parameters in Table 1, the model process uncertainty parameters, and the latent model
states and fluxes. The latent model states represented the 'true' stock or flux before measurement
uncertainty was included in the observation. Our hierarchal approach was designed to partition
uncertainty that is attributable to uncertainty in parameters, model process, and measurements
(Hobbs and Hooten, 2015). Previous forest ecosystem DA efforts have either focused on
parameter uncertainty, by using measurement uncertainty as the variance term in a Gaussian cost



function, or on total uncertainty by directly estimating the Gaussian variance term. The latter
combines measurement uncertainty and process uncertainty into the same parameter and is
unable to be used for developing prediction intervals, as prediction intervals only include
parameter and process errors (Dietze et al., 2013; Hobbs and Hooten, 2015). Here, our focus was
on estimating the probability distribution of forest biomass before uncertainty is added through
measurement.

First, we estimated the probability of a latent state or flux ($z_{i,m,p}$) for each data point (i) from each
data stream (m) in a plot (p) using the 3-PG model with the plot FR . This included the optimized
parameters ($\theta_F$), fixed parameters ($\theta_C$), soil characteristic inputs (S), climate inputs (C), site
index (SI), fertility ($FR_p$), and initial conditions (I) required by the 3-PG to simulate each plot,
$f(\theta_F, \theta_c, C, S, I, FR_p)$. The latent state ($z_{i,m,p}$) was assumed to be normally distributed with the mean
from the 3-PG simulation and an optimized, data stream-specific, process variance $\sigma^2_{m,(process)}$

p(process|process parameters)=
$P\left(z_{i,m,p} \middle| f(\theta_F, \theta_C, C, S, I, FR_p), \sigma^2_{m\ (process)}\right)$
$\sim Normal\left(z_{i,m,p} \middle| f(\theta_F, \theta_C, C, S, I, FR_p), \sigma^2_{m\ (process)}\right)$                    Equation 1

The unobserved true state related to the observed state through a data observation model. In the
sampling model, the measured state ($y_{i,m,p}$) was a random sample from a normal distribution
with a mean of the true state and a data point-specific standard deviation ($\sigma^2_{i,m,p}$).

p(data|process,data parameters)=



$\quad P\left(y_{i,m,p}\middle|z_{i,m,p},\sigma^2_{i,m,p}\right)\sim Normal\left(y_{i,m,p}\middle|z_{i,m,p},\sigma^2_{i,m,p}\right)$ $\qquad$ Equation 2

This standard deviation ($\sigma^2_{i,m,p}$) represented measurement uncertainty and was similar to the
denominator in least-squares approach that is commonly used in DA (Bloom and Williams,
2015; Keenan et al., 2011).

Each parameter ($\theta_F$) that was optimized using the Bayesian method had a prior probability that is
specified in Table 1. The prior distribution for the standard deviation $\sigma^2_{m,(process)}$ parameters
were uniformly distributed:

p(process parameters|priors)×p(priors)= $P(\sigma^2_m)\times P(\theta_F)$ $\qquad$ Equation 3

where

$P(\sigma^2_m)\sim unif(0.001,100)$ $\qquad$ Equation 4

and

$P(\theta_F)\sim$ See Supplemental Table 1 $\qquad$ Equation 5

Finally, following the description of the plot specific $FR_p$ described above, the probability for
fertilized treatments was based on a comparison to the control treatment FR.





$$P\left(FR_p|\theta_F,E\right)=\begin{cases} 1 \text{ if non-fertilized} \\ 1 \text{ if fertilized and } FR_p \geq FR \text{ of control plot} \\ 0 \text{ if fertilized and } FR_p < FR \text{ of control plot} \end{cases}$$ Equation 6

Our complete Bayesian model for estimating the posterior distributions for the parameters ($\theta_F$),
process uncertainty ($\sigma^2_{m,(process)}$), and unobserved true states ($z_{i,m,p}$) was:

$$P\left(\theta_F,\sigma^2_m,z_{i,m,p}\middle|y_{i,m,p},\sigma^2_{i,m,p},\theta_C,S,C,SI,I\right) \propto$$
$$P(z_{i,m,p}|f\left(\theta_F,\theta_C,FR_p,E\right),\sigma^2_m)P(y_{i,m,p}|z_{i,m,p},\sigma^2_{i,m,p})P\left(FR_p|\theta_F,E\right)P(\theta_F)P(\sigma^2_m)$$ Equation 7

We numerically estimated the posterior distributions using the Monte-Carlo Markov Chain –
Metropolis Hasting (MCMC-MH) algorithm (Zobitz et al., 2011). This approach has been widely
used to approximate parameter distributions in ecosystem DA research (Fox et al., 2009;
Trudinger et al., 2007; Williams et al., 2005; Zobitz et al., 2011). We adapted the size of the
jump for each parameter (i.e., how far a proposed new value can potentially be from the current
value) to ensure the acceptance rate of the parameter set is between 22% and 43% (Ziehn et al.,
2012). All MCMC-MH chains were run for 30 million iterations with the first 15 million
iterations discarded as the burn-in. Three chains were run and compared for convergence and we
sampled every 1000[th] parameter in the final 15 million iterations of the MCMC-MH chain. This
thinned chain was used in the analysis described below. The 3-PG model and MCMC-MH
algorithm were programed in FORTRAN 90 and used OpenMP to parallelize the simulation of
each plot within an iteration of the MCMC-MH algorithm.



**2.4 Model simulations**

Each plot simulated required initial conditions for each model state, climate inputs, soil

characteristic inputs, and site index. We used the first observation at the plot as the initial

conditions for the loblolly pine vegetation states (foliage biomass, stem biomass, coarse root

biomass, fine root biomass, and stem number). When observations of coarse biomass and fine

root biomass were not available, these stocks were initialized as a mean region-wide proportion

of the observed stem biomass. However, the value of initial root biomass in plots without

observations was not important because the plots without root observations did not contribute to

the root cost function and root biomass does not influence any other functions in the model. In

the two plots with flux observations (US-Dk3 and US-NC2), hardwood understory was also

initialized using the first set of observations. Initial fine root and coarse biomass was distributed

between loblolly pine and hardwoods based on their relative contribution of total initial foliage

biomass. The initialized available soil water was assumed to be equal to the maximum available

soil water because most plots were initialized in winter months when plant demand for water is

minimal. The maximum available soil water in each plot was extracted from the SSURGO soils

dataset (Staff, 2016). We assumed that the minimum available soil water was zero. The value we

used corresponded to the maximum available soil water for the top 1.5 m of the soil. Because we

focused on a region-wide optimization, we used region-wide 4-km estimates of observed

monthly meteorology as inputs and to collect the 35-year mean annual temperature for each plot

(Abatzoglou, 2013). Site index was based on height measurements at age 25 in each plot or

calculated by combining observations of height at younger ages with an empirical model



(Dieguez-Aranda et al., 2006).

We simulated the experiments by altering the environmental modifiers or by modifying the
environmental inputs. Nutrient addition experiments were simulated by directly estimating FR,
rather than calculating from Equation 2, and by requiring the optimized FR in the fertilized plot
to be equal to or greater than the FR in the control plots. Throughfall exclusion experiments were
simulated by decreasing rain inputs by 30% in the treatment plots. This assumed that the
fractional reduction in precipitation and throughfall were equal. The SETRES Irrigation
experiments were simulated by adding 650 mm to precipitation between April and October. $CO_2$
enrichment experiments were simulated by setting the atmospheric $CO_2$ input equal to the
treatment mean from the elevated $CO_2$ rings (570 ppm). While not an experiment, one plot (US-
NC2) included a thinning treatment during the period of observation. We simulated the thinning
by specifying a decrease in the stem count that matched the proportion removed at the site, with
the biomass of each tree equivalent to the average of trees in the plot.

**2.4 Model experiments and analysis**
Our analysis focused on comparing parameter distributions and predictions among simulations
that used different experimental treatments to estimate the posterior distributions (Table 4). To
examine the influence of the Duke FACE $CO_2$ fertilization, we compared a one stage vs. a two-
stage data assimilation process. The one stage process assimilated all observations in all plots
and experiments simultaneously. In this approach, the elevated $CO_2$ plots only represented 5 of
the 294 plots across the region and thus a relatively minor contribution to the likelihood (cost-
function) calculation. The two-stage process used the observations from Duke FACE, US-Dk3



flux site, the other flux site in North Carolina (US-NC2) to estimate parameter posteriors using
the priors in Table 1 and SI Table 1. These sites were grouped together because they were the
most data rich, had the high frequency data streams (monthly GEP, ET, and LAI), and were
relatively close in geography. FR was directly estimated for all plots in the first stage, with the
FR of a fertilized plot required to be equal to or higher than its control plot. The FR of the $CO_2$
experiment was equal to the corresponding control plot estimated FR. The FR of the control plot
was required to be greater than 0 and, if associated with a nutrient fertilization plot, less than the
FR of the fertilized plot.

For the second DA stage, the posterior distributions from the first stage were used as priors for
the assimilation of the region-wide observations from the PINEMAP, FPC RW 18, FMRC
Thinning, SETRES, and Waycross studies (Table 4). We compared the $CO_2$ quantum yield
enhancement parameter (Calpha700) between the one and two stage approaches to evaluate how
the estimation of $CO_2$ fertilization of plant growth depended on how the Duke FACE data are
used in data assimilation. We also estimated the distribution of the percentage increase in net
primary productivity (NPP) associated with the elevated $CO_2$ treatment using the one and two
stage data assimilation approaches. The distribution of the percentage increase in NPP was
calculated by randomly selecting 1000 parameter sets, with replacement, from the 1-stage
converged MCMC chains. This calculation was repeated using the 2-stage approach.

Based on the results from comparing the one and two stage approaches (see results below), we
proceeded using the two-stage approach to examine the influence of the water manipulation and
nutrient fertilization experiments on posterior distributions and predictions. To evaluate the



influence of water manipulation experiments, we repeated the second stage of the data
assimilation without the plots where water was added or subtracted. To evaluate the influence of
the nutrient manipulation experiments, we first repeated the first stage of data assimilation
without the nutrient addition plots in the Duke FACE experiment and used those posteriors as
priors to the second stage. This ensured that the priors to the second stage of data assimilation
did not include information from nutrient addition experiments. The second stage then excluded
the other nutrient manipulation experiments in the region.

To examine how the exclusion of the water manipulation experiments influenced parameter
inference and predictions, we first examined how the parameter distributions changed from
initial priors through the two assimilation stages. With respect to the water manipulation
experiments, we focused on the shape of the relationship between available soil water and the
quantum yield and stomatal conductance modifier (governed by parameters SW1 and SW2) with
and without assimilating the water manipulation experiments. To illustrate the capacity to
estimate the probability distribution of predictions using the posterior uncertainty in parameters,
we analyzed a focal site in Georgia, near the center of the loblolly pine range (circle in Figure 2).
At the focal site, we predicted the sensitivity of stem biomass at age 25 (hereby referred to as
$STEM_{25}$) to a 30% increase and a 30% decrease in annual precipitation with and without
assimilating the water experiments. A 30% percent decrease in precipitation mirrors the
magnitude of reduction in the experimental throughfall reduction studies used in DA (Table 3
and Figure 3). Our prediction distributions were calculated by integrating across the parameter
uncertainty by repeating simulations using 1000 random draws from the converged chain of the
posteriors. Finally, we predicted the regional response to a reduction in precipitation from



historical using the median posterior parameter values from the data assimilation with and
without the water experiments included. Our regional corresponded to the native range of
loblolly and used the HUC12 (USGS 12-digit Hydrological Unit Code) watershed as the scale of
simulation. For each HUC12 in the region we used the mean site index, 30-year mean annual
temperature, available soil water aggregated to the HUC12 level, and monthly meteorology as
inputs (Figure 2). We simulated forest development from 1989 to 2014 using actual precipitation
and again with a 30% reduction in precipitation. We focused our analysis on the percent change
in $STEM_{25}$ between the two simulations.

To examine how the exclusion of the nutrient addition experiment influenced parameter
inference and prediction, we focused on the difference in maximum quantum yield parameter ($\alpha$)
and the relationship between site index and soil fertility modifier (FR) with and without
assimilating the nutrient experiments. Additionally, we simulated how stem biomass at age 25
($STEM_{25}$) responded to a complete removal of nutrient limitation (FR = 1) for the focal site in
Georgia. As in the precipitation sensitivity described above, we represented the percentage
change in $STEM_{25}$ between simulations with estimated FR and FR =1 as a distribution by
integrating across parameter uncertainty. We predicted the regional response to nutrient
fertilization by setting the FR at all HUC12 units (see previous paragraph) equal to 1 using the
median posterior parameter values from data assimilation where nutrient addition experiments
were either included or not. We focused on the regional pattern in the percentage change in stem
biomass with the predicted FR (current level fertility) and FR = 1 (nutrient limitation removed).

Finally, we assessed overall model performance of the 2-stage approach for data assimilation



with all experimental types included in DA, excluding the nutrient addition experiments, and
excluding the nutrient addition experiments using an out-of-sample approach. The approach held
40 random FMRC thinning study plots (Table 3) out from the assimilation, predicted the 40 plots
using the median parameter values, and compared the predicted stem biomass to the observed
stem biomass. These were plots without any manipulations of nutrients or water, were located
throughout the region, and had measurement ages up to 30 years old. For each plot, we only used
the most recent observed values to increase the time length between initialization and validation.
We repeated the validation for four unique sets of 40 FMRC thinning study plots.

**3    Results**
Our multi-site, multi-experiment, multi-data stream DA approach was able to constrain most
parameters in the 3-PG model (31 of 46 parameters were optimized; Table 6; Supplemental
Table 3; Supplemental Figure 1-3). The 31 optimized parameters were the most sensitive
parameters in the 3-PG model, defined by the change in total biomass at age 25 for the focal site
in Georgia to a 10% change in the parameter (Table 1; Supplemental Table 1). One exception
was the light extinction coefficient (k), which showed high sensitivity but was assumed to be
fixed because it strongly co-varied with the quantum yield parameter ($\alpha$). Parameters associated
with biomass allocation had priors with large variance but DA was able to provide posteriors
with relatively low variance (pFS2, pFS20, pR, and pCRS; Supplemental Figure 1; Supplemental
Table 3). The DA process also produced posterior distributions that had less variability than the
prior distribution for the important parameters associated with light-use efficiency ($\alpha$, y, FR1,
and FR2; Table 5). DA did not change the parameter distributions, i.e., the posterior and prior
distributions were similar, for the parameters that governed the temperature sensitivity of





quantum yield, the VPD sensitivity of quantum yield, and the maximum canopy conductance
(Supplemental Figure 1-2; Supplemental Table 3). These parameters had strong priors supported
by previous research on loblolly pine physiology. Finally, the DA approach was able to estimate
the distributions of the process uncertainty parameters (Supplemental Figure 3; Supplement
Table 4).

The addition of the second stage of assimilation that used region-wide observations and
posteriors from the DK+NC2 assimilation modified the distributions of the parameters that
related to allocation and mortality but did not provide additional constraint on the physiological
parameters (Table 5). In particular, the parameters associated with the self-thinning curve and
allocation of coarse roots had non-overlapping 95% credible intervals between the DK+NC2 and
RW assimilation. The larger estimate for Wsx1000 and lower value for thinPower in the
DK+NC2 indicated self-thinning was lower at the sites in the DK+NC2 assimilation than the
average of the other sites in the region. The lower value for the pCRS parameter indicated that
less NPP was allocated to coarse roots in the DK+NC2 assimilation than the RW assimilation.

The two-stage assimilation was critical for constraining the $CO_2$ quantum yield enhancement
parameter (Calpha700). Both the mean of the posterior distribution and the range of the 95%
credible interval were smaller for fCalpha700 when all observations were assimilated
simultaneously (1-stage approach) than the distribution estimated using the 2-stage approach
(Duke and NC2 assimilated before the region-wide assimilation) (Figure 5a; Table 5). Despite
the same data used in both approaches, the differences in fCalpha700 led to a predicted lower
enhancement of NPP associated with elevated $CO_2$ in the experiment. The 1-stage assimilation





approach had a median increase in NPP between the control and elevated $CO_2$ treatments of 15%
compared to a 27% in the two-stage approach (Figure 5b).

The RW assimilation constrained the soil fertility parameters that were necessary to enable
regional simulations. Our regional model using the 2-stage approach performed well compared
to stem biomass data not used in the assimilation. The mean bias in stem biomass of the four out-
of-sample validation sets was -6.7 % and the RMSE was 21.2 Mg ha$^{-1}$ (Figure 4).

Excluding the nutrient addition experiments from the DA increased the simulated level of
nutrient limitation but did not change the predictive capacity of the independent non-manipulated
validation set. DA without nutrient fertilization experiments had a greater and more uncertain
value for the maximum quantum yield parameter ($\alpha$; Figure 6a; Table 5). This parameter was
shared across all plots and modified by the environmental conditions at each plot. To compensate
for the higher $\alpha$ parameter when nutrient fertilization experiments were excluded from DA, the
two soil fertility parameters (FR1 and FR2) combined to predict a 10% lower FR values for a
given site index and mean annual temperature (Figure 6b). Subsequently, the prediction for the
percentage change in STEM$_{25}$ associated with maximum fertilization (i.e., setting FR = 1) at the
focal site in Georgia was 7% higher and had greater uncertainty when nutrient fertilization
experiments were excluded from the DA (Figure 6c). The RMSE and mean bias of the non-
manipulated validation set was 20.4 Mg ha$^{-1}$ and -4.8 %, respectively (SI Figure 1a)

Excluding the water manipulation experiments from the DA reduced the sensitivity to available
soil water but, similar to the inclusion of the nutrient addition experiments, did not change the



predictive capacity of the independent non-manipulated validation set. The combined differences
in the SW1 and SW2 parameters between the DA with and without the water manipulation
experiments decreased the sensitivity of quantum yield and canopy conductance to a reduction in
available soil water (Figure 7a). For example, at an available soil water to maximum available
soil water ratio of 0.50, the quantum yield and canopy conductance modifier decreased from 0.95
without water experiments to 0.8 with water experiments (Figure 7a). At the focal site in
Georgia, the sensitivity of $STEM_{25}$ to a reduction in annual precipitation (Figure 7b) was larger
when the water experiments were included in the DA (-8.5% median change in $STEM_{25}$ for a
30% reduction in precipitation) than when the experiments were excluded (-4.1% median change
in $STEM_{25}$ for a 30% reduction in precipitation). Similarly, the predictions of $STEM_{25}$ change
associated with a 30% increase in precipitation (median: 3.8%) were higher when water
experiments were included than when not included (median: 1.1%). The magnitude of
uncertainty in the predictions did not differ substantially between forecasts with and without
water experiments (Figure 7b). The RMSE and mean bias of the non-manipulated validation set
was 19.3 Mg ha$^{-1}$ and -5.8 %, respectively (SI Figure 1b)

Regionally (i.e., the native range of loblolly pines), using the two-stage approach (RW), the most
productive areas were the coastal plains and the interior of Mississippi and Alabama (Figure 8).
These patterns were largely driven by patterns in the soil fertility factor (FR; Figure 9), reflecting
the sensitivity of the 3-PG model to the FR parameters (Table 1). The area weighted mean
$STEM_{25}$ response to fertilization (represented by setting FR = 1) across the region was 28% with
the highest response occurring in the far west of the region, the Piedmont of Georgia, the interior
of the gulf coast, and the northern reach of the region (Figure 10a). These were all areas with the



lowest soil fertility parameter. The least responsive region to nutrient addition was in Florida
(Figure 10a). Excluding the nutrient addition experiments from the DA increased the sensitivity
to nutrient addition (Figure 10b), as shown for the focal Georgia site (Figure 6b), but did not
change the spatial patterns of the response.

The sensitivity of forest production to a 30% reduction in precipitation varied across the region.
The most sensitive areas, the Piedmont of Georgia and the western edge of the region, predicted
up to a 13.1% decline in $STEM_{25}$ (Figure 11a). These were warm areas with relatively low
precipitation before the 30% reduction (Figure 2c). The least sensitive area was the interior of the
gulf coast (<1% decline; Figure 11a), the area with the highest precipitation in the region (Figure
2c). The regional mean reduction in $STEM_{25}$ associated with a 30% decrease in precipitation was
5.7% (Figure 11a). Excluding the water manipulation experiments from DA reduced the regional
mean sensitivity to 1.7% (Figure 11b).

**4    Discussion**

Using DA to parameterize models applied to forecasting ecosystem change requires detangling
the vegetation responses to temperature, precipitation, nutrients, and elevated $CO_2$. To address
this challenge, we introduced a regional-scale hierarchical Bayesian approach (DAPPER) that
assimilated data across environmental gradients and ecosystem manipulation experiments into a
modified version of the 3-PG model to estimate parameters and generate uncertainty estimates
on predictions of carbon and water cycling across the whole native range of loblolly pine.
Furthermore, we organized observations of carbon stocks, carbon fluxes, water fluxes, vegetation



structure, and vegetation dynamics that spanned 35 years of forest research (Figure 3; Table 3) in
a region with large and dynamic carbon fluxes (Lu et al., 2015). By combining the DAPPER
system with the regional set of observations, we were able to estimate parameters in a model
with high predictive capacity (Figure 4) and with quantified uncertainty on parameters (Table 5).
We also found that the predictions of forest productivity response to rising $CO_2$, altered
precipitation, and altered nutrient availability were highly sensitive to the types of experiments
used in DA as well as the methodological approach applied.

We found that including nutrient and water manipulation experiments aided in distinguishing the
mechanisms driving patterns in biomass across the region. Including these experiments in the
data-assimilation did not improve the predictive capacity of the independent validation set of
non-manipulation plots. However, including nutrient and water manipulation did change the
underlying mechanisms explaining the patterns in stem biomass. Without the nutrient and water
manipulation experiments, the same biomass predictions were attributable to a higher level of
nutrient limitation and a lower level of water limitation. This resulted in differing sensitivities to
changes in nutrient or water availability.  Overall, this finding highlights a key challenge when
parameterizing ecosystem models that will be used for global change predictions, that different
combinations of environmental drivers can produce similar predictions of current observations.
Ecosystem manipulation experiments are an important tool for addressing this challenge.

Parameter and process identifiability, or equifinality, presents a challenge when parameterizing
ecosystem models using DA (Luo et al., 2009). One important source of equifinality is the
tradeoff between parameters governing the potential productivity of the vegetation and the





downregulation of productivity due to nutrient limitation. When using observational data at a
single site, a single parameter is often optimized to set a photosynthetic rate per absorbed light,
i.e., a quantum yield. This single parameter combines the potential photosynthesis set by climate
and the influence of nutrient limitation on photosynthesis into a single parameter. However,
separating these two processes into two or more parameters is challenging because a high
potential quantum yield parameter ($\alpha$) and high nutrient limitation (FR) can mathematically yield
the same photosynthetic rate as low potential quantum yield and low nutrient limitation. The
former implies a larger potential response to nutrient addition than the latter. We found that
including nutrient addition experiments in DA helped overcome this challenge. In the case of the
3-PG model used in this study, the maximum quantum parameter ($\alpha$) and soil fertility parameters
(FR1 and FR2) were more constrained and inferred lower levels of nutrient limitation across the
region when nutrient fertilization experiments were included in the DA. This finding likely
extends to other models that include the concept of potential productivity and productivity
downregulated by nutrient limitation. For example, the applications of the Data Assimilation
Link Ecosystem Carbon (DALEC) model (Williams et al., 2005) to DA often assumed nine of
the ten parameters associated with photosynthesis were fixed, thus using a single parameter to
represent both the quantum yield (defined as nitrogen use efficiency in DALEC) and the
magnitude of nitrogen limitation of a site (Fox et al., 2009). The use of a single parameter, rather
than using nutrient addition experiments to separate into multiple parameters, is appropriate
when assuming nutrient availability is static. Applications of DA to predictions of ecosystems
with changing nutrient availability, either through management, elevated $CO_2$, or nitrogen
addition, would benefit from using nutrient addition studies to quantify the magnitude of nutrient
limitation. Studies of known nutrient gradients could be used in lieu of nutrient addition studies,





but effort must be made to account for confounding abiotic factors, such as available soil water
or climatic conditions, that may co-vary with nutrient availability.

Another challenge in DA is deciding how to weigh different types of data used in model fitting
(Gao et al., 2011; Wutzler and Carvalhais, 2014). Here we demonstrate that DA efforts should
also consider how to weigh different types of ecosystem experiments. In our analysis, we
included three types of experiments: nutrient addition, water manipulation, and $CO_2$ fertilization.
The nutrient addition and water manipulation experiments were represented by multiple sites
across the region while the $CO_2$ fertilization only occurred at a single location (Figure 3). We
found that the parameter that represents the increase in maximum quantum yield under elevated
$CO_2$ was substantially lower when all observations, sites, and experiments were assimilated
simultaneously than when the $CO_2$ fertilization experiment was given greater weight. The greater
weight was applied by first assimilating the $CO_2$ fertilization experiment and using the posteriors
as priors for assimilating the remaining observations. Providing additional weight on the single
site with unique environmental conditions (i.e., atmospheric $CO_2$ at 570 ppm) using a two-stage
data-assimilation, we were able to more accurately represent the observed differences in NPP
between the ambient and elevated $CO_2$ treatments at the Duke site (McCarthy et al., 2010).
Given than only a few of the parameters were significantly different between the Duke site and
the other studies across the region, it may be possible to optimize one parameter for the Duke site
and another parameter for the other studies in a 1-stage approach that combines all the plots into
a single assimilation. However, the 2-stage approach was required to identify which parameters
were different between the Duke site and the other studies. Overall, we suggest that DA efforts
using multiple studies and multiple experiment types identify whether particular experiments at



limited number of sites have the potential to uniquely constrain specific parameters. In this case,
additional weight may be needed to avoid having the signal of the unique experiment
overwhelmed by the large amount of data from the other sites and experiments.

Our analysis highlights that nutrient limitation of productivity was widespread across the region.
The largest potential gains in productivity from nutrient addition were predicted in central
Georgia, an area with warm annual temperatures but poor soils, as expressed in the low site
index. The baseline fertility used in our regional analysis was derived from an empirical model
of site index that was developed using field plots with minimal management (Sabatia and
Burkhart, 2014). Subsequently our estimate of baseline fertility is likely on the low end of forest
stands currently in production. Further, we recognize that the site index model had uncertainty
that could be formally incorporated into the hierarchal Bayesian approach in future applications.

The soil fertility modifier has commonly been used to calibrate the 3-PG for applications to a
single site, with recent work focused on developing an approach to predicting the soil fertility
modifier from environmental conditions (Gonzalez-Benecke et al., 2016; Subedi et al., 2015).
We have extended prior efforts to develop a simple predictive model of FR in two ways. First,
we simultaneously calibrated the parameters in the empirical FR model alongside the other
parameters in the 3-PG model. Prior studies have assumed fixed values for the 3-PG model
parameters, fitted FR for plots with observations, and developed a relationship between FR and
site index. Our Bayesian approach to simultaneously calibrating the 3-PG parameters and the FR
model allowed for the estimation of uncertainty and covariation among parameters in the 3-PG
and FR models. Second, we included a climate term (mean annual temperature) in the



relationship between site index and FR. This resulted in a lower FR for a given site index in
warmer locations. By including the climate term, FR can be interpreted as relative to the climate
at a given location and the potential productivity of a plot can be optimized by setting FR equal
to 1. When a climate term is not used in the empirical FR model, FR is relative to the greatest
site index in the region, which does not occur in the northern extent of the region even in
fertilized plots due to climatic constraints.

Our simulations show that loblolly pine productivity was not strongly sensitive to changes in
precipitation at present day temperatures and atmospheric $CO_2$. We simulated a 30% reduction in
annual precipitation and found a maximum of a 13.1% reduction in productivity. A 30%
reduction in precipitation is plausible but is more extreme than most Multivariate Adaptive
Constructed Analogs (MACA) downscaled climate model projections for the Representative
Concentration Pathway (RCP) 8.5 scenario from the CMIP5 Project (comparing the 1971-2000
period to the 2070-2099) (Abatzoglou and Brown, 2012; Taylor et al., 2012). Central Georgia
was the most responsive to precipitation reduction, paralleling the spatial patterns in the response
to nutrient addition, suggesting that the region is able to support high productivity but is sensitive
to nutrient and precipitation levels. The simulated sensitivity was likely due to poor soils (low
site index) and low baseline precipitation relative to the warm climate. Our predictions of low
sensitivity to precipitation reduction or addition were derived from assimilating observations
from throughfall exclusion and irrigation experiments across the region. Prior publications from
the studies used in DA also reported low sensitivities to water manipulations, indicating that our
predictions are likely not biased (Albaugh et al., 2004; Samuelson et al., 2014; Ward et al., 2015;
Wightman et al., 2016). For example, the throughfall exclusion experiment at the focal site in





Georgia, reported a 13% reduction in stem production during a dry year but a 0% reduced during
a wet year, resulting in a 7% reduction of productivity over a 2-year period in response to a 30%
reduction in throughfall (Samuelson et al., 2014). Our predicted 8.5% reduction to a 30%
reduction in precipitation compares well to the observed change, noting that our sensitivity
integrated over a 25-year rotation and included a mix of relatively wet and dry years.

The 3-PG model included a highly simplified representation of interactions between the water
and carbon cycles that resulted in parameterizations that, while consistent with observations, may
contain assumptions that require additional investigation. For example, transpiration is modeled
as a potential canopy transpiration that occurred if leaf area was not limiting transpiration. The
LAI at which leaf area was no longer limiting was a parameter that was optimized (LAIgcx in SI
Table 3), resulting in a value of 2.3.  Interestingly, this optimized value is consistent with the
scant literature on this topic.  In their analysis of multi-year measurements of transpiration in
loblolly pine, Phillips and Oren (2001) observed that transpiration per unit leaf area was
relatively insensitive to increases in leaf area above LAI of approximately 2.5.  Iritz and Lindroth
(1996) reviewed transpiration data from a range of crop species and found only small increases
in transpiration above LAI of 3-4.  These authors suggest that the threshold-type responses
observed were related to the range of LAI at which self-shading increases most rapidly, therefore
limiting increases in transpiration.  The resulting model behavior of "flat" transpiration above 2.3
LAI, with gradually decreasing photosynthesis above that value, results in increasing water use
efficiency at higher LAI values.  The parameterization of the relationships between transpiration
and photosynthesis in 3-PG would likely benefit fromadditional data beyond the two eddy-
covariance studies with ET observations used here. For example, canopy conductance estimates,





and their associated uncertainty have been derived from assimilating observations from sap-flow
measurements into a model that scales from the sensor measurements to canopy transpiration
using LAI observations (Bell et al., 2015). This sap-flow to canopy conductance scaling
approach (the State Space Canopy Conductance (StaCC) model (Bell et al., 2015)) produces a
probability distribution of monthly canopy conductance that could be integrated into the
DAPPER system by treating the posterior estimates of StaCC as the distribution of the data in
equation 2. Second, the optimized parameters that described the relationship between relative
available soil water and the modifier of photosynthesis and transpiration predicted a modifier
value greater than zero when the relative available soil water was zero. This resulted in positive
values from photosynthesis and transpiration when the average available soil water during the
month was zero. In practice, the monthly available soil water was rarely zero during simulations,
which presents a challenge constraining the shape of the available soil water modifier. The priors
for the two available soil moisture modifiers (SW1 and SW2) had ranges that permitted the
modifier to be zero. Therefore, additional data is likely needed during very dry conditions to
develop a more physically based parameterization. Alternatively, the parameterization of a non-
zero soil moisture modifier at zero available soil water may be due to trees having access to
water at soil depths deeper than the top 1.5 m of soil represented by the bucket in 3-PG. Overall,
it is important to view the parameterization presented here as a phenomenological relationship
that is consistent with observations from throughfall exclusion and irrigation experiments as well
as observations across regional gradients in precipitation.

Beyond the specifics of the 3-PG modeling efforts, the DA of regional observations into a
monthly, computationally tractable ecosystem model can potentially inform Earth system





modeling efforts. While the details of physiology differ between 3-PG and global land-surface
models, the concepts governing NPP allocation are similar. Therefore, DA using the 3-PG model
can be used to parameterize the allocation patterns of similar plant types in a global model. One
land-surface model, the Community Land Model (CLM), includes parameters that govern the
ratio of stem to leaf allocation, ratio of coarse root to stem allocation, and the ratio of leaf to fine
root allocation, parameters that are also optimized in DAPPER. As an example, the ratio of fine
root to leaf allocation in CLM 4.0 and 4.5 for temperate pine plant function type is set to 1,
resulting in equal annual allocation of carbon to foliage and fine roots (Oleson et al., 2013). In
contrast, we found that the median ratio of fine root to foliage allocation was substantially lower
at 0.13 (Table 6). Therefore, simulations in the CLM with the lower value of root allocation
would have higher allocation to aboveground tissues if the loblolly pine parameters from our
analysis were used. This would increase carbon accumulation in woody tissues and could alter
predictions of nutrient limitation because stems have higher C:N ratios. Other parameters,
including the stem to coarse root ratio, are closer to the values used in the CLM.

## 5 Conclusions

DA is increasingly used for ecological forecasting due to its ability represent prior knowledge,
integrate observations into the parameterization, and estimate multiple components of
uncertainty, including observation, parameter, and process representation uncertainty (Dietze et
al., 2013; Luo et al., 2011b; Niu et al., 2014). Our application of DA to loblolly pine plantations
of the southeastern U.S demonstrated that these ecosystems are well suited as a test-bed for the
development of DA techniques, particularly techniques for assimilating ecosystem experiments.
Further, we found that assimilating ecosystem manipulative experiments into a simple ecosystem



model changed predictions quantifying how forest productivity responds to environmental
change, highlighting the importance of networks of ecosystem manipulation experiments for
helping to parameterize and evaluate ecosystems models (Medlyn et al., 2015).

**6 Data availability**
Observations used in the DA can be found in the following: Duke FACE study can be found in
McCarthy et al. (McCarthy et al., 2010), the PINEMAP studies are available through the TerraC
database (http://terrac.ifas.ufl.edu), the DK3 eddy-flux tower data are available through the
Ameriflux database (http://ameriflux-data.lbl.gov) , the Waycross data can be found in Bryars et
al. (2003), the NC2 data are available upon request with Asko Noormets, the FMRC and FPC are
available through membership with the cooperatives. The parameter chains and 3-PG are
available upon request from R. Quinn Thomas.

**Acknowledgments**
Funding support came from USDA-NIFA Project 2015-67003-23485 and the Pine Integrated
Network: Education, Mitigation, and Adaptation project (PINEMAP), a Coordinated
Agricultural Project funded by the USDA National Institute of Food and Agriculture, Award
#2011-68002-30185. Additional funding support came from USDA-NIFA McIntire-Stennis
Program. The Virginia Space Grant Consortium Graduate STEM Research Fellowship Program
provided partial support for A. Jersild. Computational support was provided by Virginia Tech
Advanced Research Computing. This research was also supported by grants from the French
Research Agency (MACACC ANR-13-AGRO-0005 and MARIS ANR-14-CE03-0007). We
thank Luke Smallman and Mat Williams for helpful discussions about data assimilation. We



thank the corporate and government agency members of the FPC and FMRC research
cooperatives for supporting the extensive long-term experimental and observational plots in
those datasets.

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





Table 1. A subset of parameters optimized using data assimilation, prior distributions, and the sensitivity of total biomass at age 25 to the parameter. These are the parameters referred to in the results and discussion, other optimized model parameter can be found in the supplemental material.

| Parameter | Parameter description | Units | Sensitivity* | Prior distribution | Prior parameters | Reference for prior |
|---|---|---|---|---|---|---|
| $\alpha$ | Canopy quantum efficiency (pines) | mol C mol PAR$^{-1}$ | 0.84 | Uniform | Min = 0.02 Max = 0.1 | Vague |
| y | Ratio NPP/GPP | - | 0.84 | Uniform | Max= 0.66 Min = 0.30 | 1 |
| fCalpha700 | Proportional increase in canopy quantum efficiency between 350 and 700 ppm CO2 | - | 0.08 | Uniform | Min = 1.05 Max = 2.0 | Vague |
| fCpFS700 | Proportional decrease in allocation to foliage between 350 and 700 ppm CO2 | - | 0.00# | Uniform | Min = 0.50 Max = 1.00 | Vague |
| SWconst | Moisture ratio deficit when downregulation is 0.5 | - | 0.06 | Uniform | Min = 0.6 Max = 1.8 | 2, Vague |
| SWpower | Power of moisture ratio deficit | - | 0.06 | Uniform | Min = 1 Max= 13 | 2, Vague |
| FR1 | Fertility rating parameter 1 (mean annual temperature coefficient) | - | 0.23 | Uniform | Min = 0.0 Max = 1.0 | Vague |
| FR2 | Fertility rating parameter 2 (site index age 25 coefficient) | - | 0.39 | Uniform | Min = 0.0 Max = 1.0 | Vague |
| wSx1000 | Maximum stem mass per tree at 1000 trees/ha | kg tree$^{-1}$ | 0.43 | Normal | Mean = 235 Sd = 25 | 3,4 |
| thinPower | Power in self thinning law | - | 0.25 | Uniform | Min = 1.1 Max = 1.80 | 3,4 |
| pCRS | Ratio of coarse roots to stem allocation | - | 0.08 | Uniform | Min = 0.15 Max = 0.35 | 5 |

1(DeLucia et al., 2007);²(Landsberg and Waring, 1997), ³(Bryars et al., 2013),⁴(Gonzalez-Benecke et al., 2016), 5(Albaugh et al., 2005)
* Sensitivity is 1 when a 10% increase in the parameter results in a 10% change in total biomass. #Sensitivity is 0 when a 10% increase in the
parameters does not change total biomass by a value greater than 0.01%.




Table 2. Regional observational data streams used in data assimilation.

| Data stream | Measurement frequency | Measurement or estimation technique | Uncertainty | Stream ID for Table 4 |
|---|---|---|---|---|
| Foliage biomass (Pine) | Annual or less | Allometric relationship | Based on propagating the allometric model uncertainty in Gonzalez-Benecke et al. 2014. Varied by observation. | 1 |
| Foliage biomass (hardwood) | Annual or less | Allometric relationship | Assumed zero | 2 |
| Stem biomass (pine) | Annual or less | Allometric relationship | Based on propagating the allometric model uncertainty in Gonzalez-Benecke et al. 2014. Varied by observation. | 3 |
| Stem biomass (hardwood) | Annual or less | Allometric relationship | Assumed zero | 4 |
| Coarse root biomass (combined) | Annual or less | Allometric relationship | Standard deviation (SD) = 10% of observation | 5 |
| Fine root biomass (combined) | Annual or less | Allometric relationship | SD = 10% of observation | 6 |
| Foliage biomass turnover (combined) | Annual | Litterfall traps | SD = 2.5% of observation | 7 |
| Fine root biomass turnover (combined) | Annual | Mini-rhizotrons | SD = 10% of observation | 8 |
| Pine stem count | Annual or less | Counting individuals | 1% (assumed small) | 9 |
| Leaf area index (pine) | Monthly to annual | Litter traps or LI 2000 | If litter trap method: SD = 2.5% of observation If LI-2000 method: SD = 10% of observation | 10 |
| Leaf area index (hardwood) | Monthly to annual | Litter traps or LI 2000 | If litter trap method: SD = 2.5% of observation If LI-2000 method: SD = 10% of observation | 11 |
| Leaf area index (combined) | Only used if not separated into pine and hardwood | Litter traps or LI 2000 | If litter trap method: SD = 2.5% of observation If LI-2000 method: SD = 10% of observation | 12 |
| Gross Ecosystem Production | Monthly | Modeled from flux eddy-covariance net ecosystem exchange | SD = 10% of observation | 13 |
| Evapotranspiration | Monthly | Eddy-covariance | SD = 10% of observation | 14 |






Table 3. Descriptions of the studies used in data assimilation.

| Study name | Number of locations | Number of plots per site | Experimental treatments (plots) | Data streams (Table 2) | Measurement Years | Measurement Stand Ages (years) | Reference |
|---|---|---|---|---|---|---|---|
| FMRC[1] Thinning Study | 163 | 1 | None | 1, 3,9 | 1981 - 2003 | 8 - 30 | (Burkhart et al., 1985) |
| FPC[2] Region-wide 18 | 18 | 2 | Nutrient addition | 1, 3,9 | 2011-2014 | 12-21 | (Albaugh et al., 2015) |
| PINEMAP[3] | 4 | 16 | Nutrient addition, 30% throughfall, Nutrient x throughfall | 1, 3,9 | 2011-2015 | 3 – 13 | (Will et al., 2015) |
| Waycross | 1 | 2 | Nutrient addition | 3,9,10 | 1991-2010 | 4-23 | (Bryars et al., 2013) |
| SETRES[4] | 1 | 16 | Nutrient addition, irrigation, nutrient x irrigation | 1,3,5,6,9, 10 | 1991-2006 | 8 - 23 | (Albaugh et al., 2004) |
| Duke FACE[5] and flux | 1 | 12 | $CO_2$, nutrient addition, $CO_2$ x nutrient addition | 2,3,4,5,6, 7,8,9,10, 11,13,14 | 1996-2004 | 13-22 | (McCarthy et al., 2010; Novick et al., 2015) |
| NC2 Flux | 1 | 1 | None | 2,3,4,5,6, 7,9,10,11 ,12,13,14 | 2005-2014 | 12-22 | (Noormets et al., 2010) |
| Total | 187 | 294 | | | 1981 - 2014 | 4 - 30 | |

[1]Forest Modeling Research Cooperative; [2] Forest Productivity Cooperative; [3] Pine Integrated Network: Education,
Mitigation, and Adaptation project (PINEMAP); [4] Southeast Tree Research and Education Site; [5] Free Air Carbon
Enrichment





Table 4. Description of the different data assimilation approaches used.

| Simulation Name | Treatments included in assimilation | Number of plots |
|---|---|---|
| All | 1-stage data assimilation. All plots and experiments in the region were used simultaneously. | 294 |
| DK+NC2 | $1^{st}$ stage of 2-stage assimilation. All plots at the Duke eddy flux (DK3), Duke Free Air $CO_2$ Enrichment Study, and NC2 eddy flux site; includes $CO_2$ enrichment and nutrient addition experiments at the Duke site | 13 |
| DK+NC2-fert | $1^{st}$ stage of 2-stage assimilation. Same as DK+NC2 but without nutrient fertilization plots | 10 |
| RW | $2^{nd}$ stage of 2-stage assimilation. Region-wide assimilation of FRMC, FPC, PINEMAP, Waycross, and SETRES sites. Uses the posteriors of the DK+NC2 simulation as priors. Includes nutrient addition and water manipulation experiments. This simulation is repeated four times for four different out-of-sample validation plots. | 281 |
| RW-fert | $2^{nd}$ stage of 2-stage assimilation. Same as RW but without nutrient addition experiments; uses the posteriors of the DK+NC2-fert simulation as priors | 222 |
| RW-water | $2^{nd}$ stage of 2-stage assimilation. Same as RW but without water manipulation experiments | 241 |




Table 5. Posterior means and 95% credible intervals for parameters listed in Table 1 using the data assimilation approaches listed in Table 4.

| Parameter | RW | RW-fert | RW-water | DK+NC2 | All |
|---|---|---|---|---|---|
| $\alpha$ | 0.037 | 0.040 | 0.037 | 0.035 | 0.032 |
|  | (0.034 – 0.040) | (0.036 – 0.045) | (0.035 – 0.040) | (0.030 – 0.042) | (0.030 – 0.035) |
| y | 0.48 | 0.48 | 0.48 | 0.48 | 0.52 |
|  | (0.46 – 0.51) | (0.45-0.51) | (0.46 – 0.51) | (0.45 – 0.51) | (0.50 – 0.54) |
| fCalpha700 | 1.31 | 1.31 | 1.31 | 1.32 | 1.11 |
|  | (1.22 – 1.40) | (1.22 – 1.40) | (1.22 – 1.40) | (1.23 – 1.41) | (1.08 – 1.15) |
| fCpFS700 | 0.84 | 0.83 | 0.84 | 0.84 | 0.99 |
|  | (0.75 – 0.93) | (0.75 – 0.93) | (0.75 – 0.93) | (0.76-0.93) | (0.95 – 1.0) |
| SWconst | 1.48 | 1.31 | 1.8 | 1.30 | 1.57 |
|  | (1.09 – 1.85) | (0.95 – 1.70) | (1.47 – 2.15) | (0.89 – 1.76) | (1.08 – 1.79) |
| SWpower | 1.61 | 1.29 | 2.93 | 2.20 | 1.47 |
|  | (0.90 – 2.46) | (0.78 – 1.98) | (1.48 – 3.82) | (1.47 – 3.44) | (1.09 – 2.26) |
| FR1 | 0.094 | 0.096 | 0.118 | not fit | 0.094 |
|  | (0.086 – 0.104) | (0.088 – 0.103) | (0.110 – 0.128) |  | (0.087 – 0.102) |
| FR2 | 0.144 | 0.124 | 0.179 | not fit | 0.153 |
|  | (0.133 – 0.154) | (0.108 – 0.142) | (0.156 – 0.182) |  | (0.140 – 0.168) |
| wSx1000 | 176 | 180 | 180 | 258 | 181 |
|  | (171 – 181) | (174 – 186) | (176 – 186) | (228 – 295) | (174 0 187) |
| thinPower | 1.67 | 1.70 | 1.71 | 1.28 | 1.61 |
|  | (1.60 – 1.74) | (1.63 – 1.78) | (1.65 – 1.78) | (1.12 – 1.60) | (1.51 – 1.69) |
| pCRS | 0.26 | 0.24 | 0.25 | 0.17 | 0.28 |
|  | (0.25 – 0.27) | (0.23 – 0.25) | (0.24 – 0.26) | (0.16 – 0.19) | (0.27 – 0.29) |




Figures

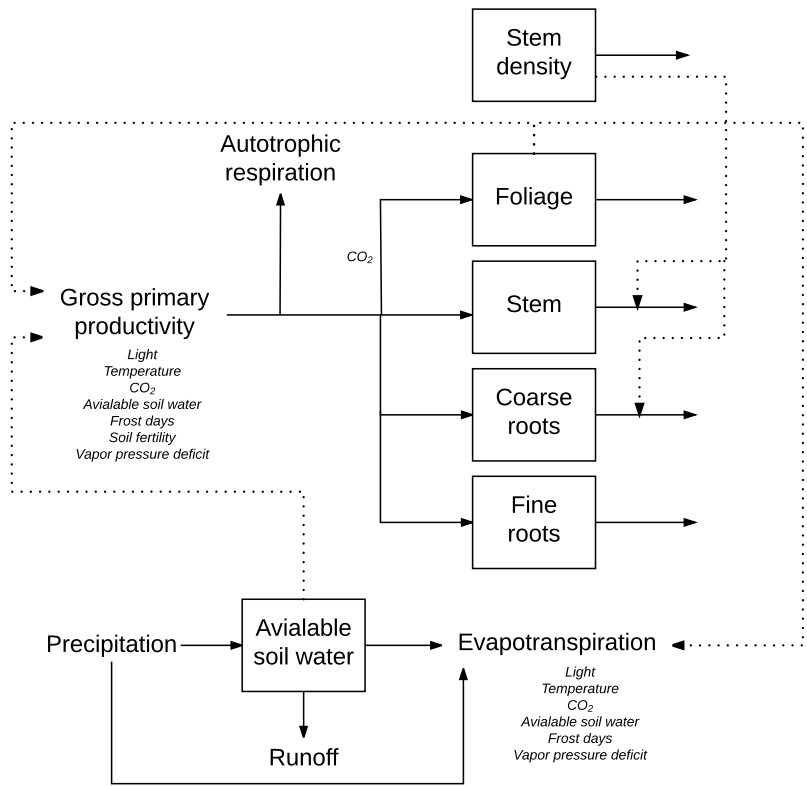

Figure 1. A diagram of the monthly time-step 3-PG model used in this study. The stocks are represented by the
boxes and the fluxes by the arrows. An influence of a stock on a flux that is not directly related to that stock is
represented by the dotted lines. The environmental influences on a flux is described using italics. A description of
the model can be found in the supplemental information.



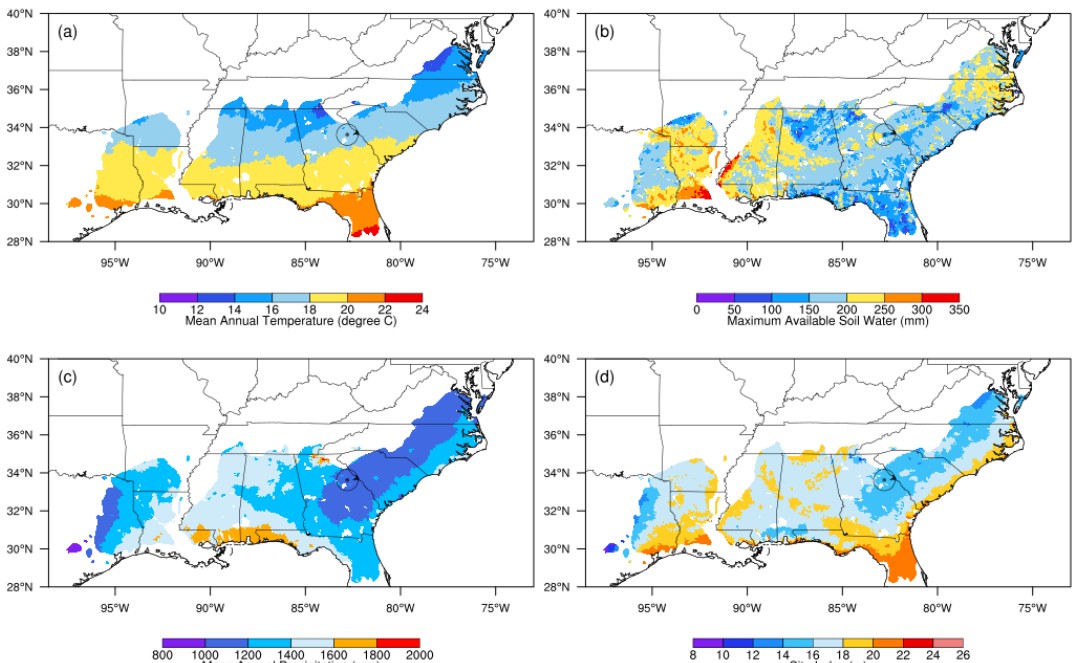

Figure 2. Key climatic and stand characteristic inputs to the regional 3-PG simulations: (a) Mean
annual temperature (1979-2011) as a summary of the gradient in monthly temperature inputs
used in simulations, (b) maximum available soil water for the top 1.5 meters of soil from
SSURGO, (c) mean annual precipitation (1979-2011) as a summary of the gradient in monthly
precipitation inputs used in simulations, and (d) site index. The focal site in Georgia highlighted
in Figures 5c and 6b is represented by the circle containing the dot. The area shown is the natural
range of loblolly pine (*Pinus taeda L.*).





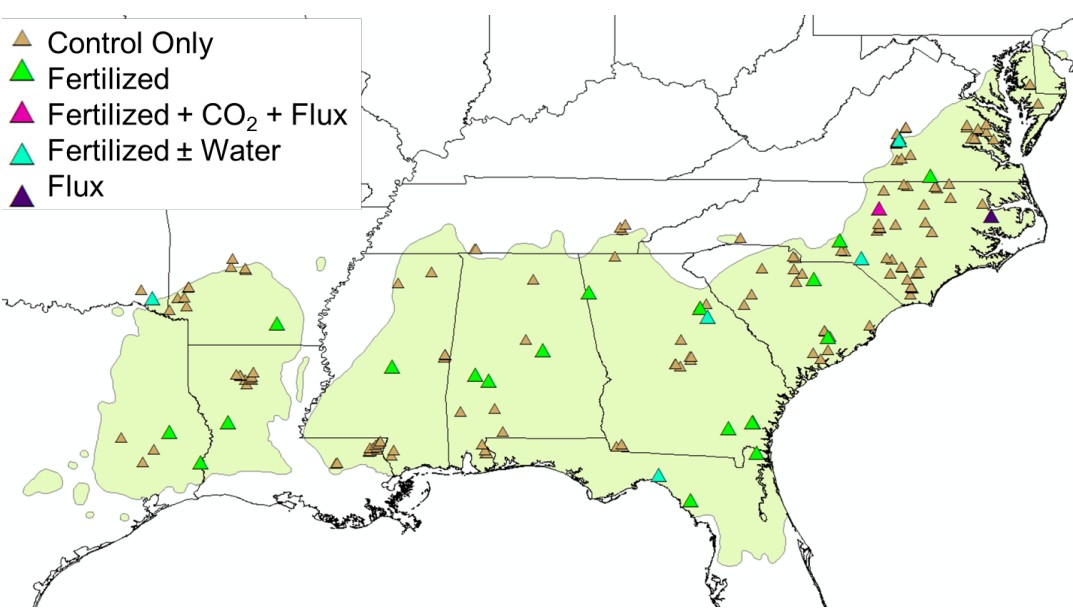

Figure 3. Map of loblolly pine distribution, plot locations used in data assimilation, and the
experiment type associated with each plot. The control-only treatments were plots without any
associated experimental treatment or flux measurements. Fertilized were plots with nutrient
additions. $CO_2$ were plots with free-air concentration enrichment treatments. The flux treatments
were plots with eddy-covariance measurements of ecosystem-scale carbon and water exchange.
The water treatments included throughfall exclusion and irrigation experiments.





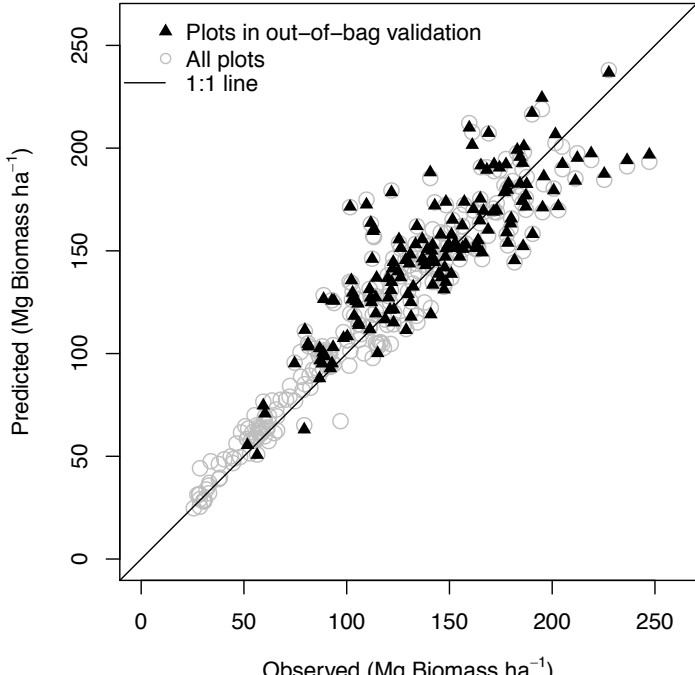

Figure 4. Model evaluation of stem biomass using the RW simulation described in Table 5. The
gray circles correspond to predictions where all plots were used in data assimilation. The black
triangles correspond to predictions where 120 plots were not included in data assimilation and
represent an independent evaluation of model predictions (out-of-bag validation). For each plot,
we used the measurement with the longest interval between initialization and measurement for
evaluation.





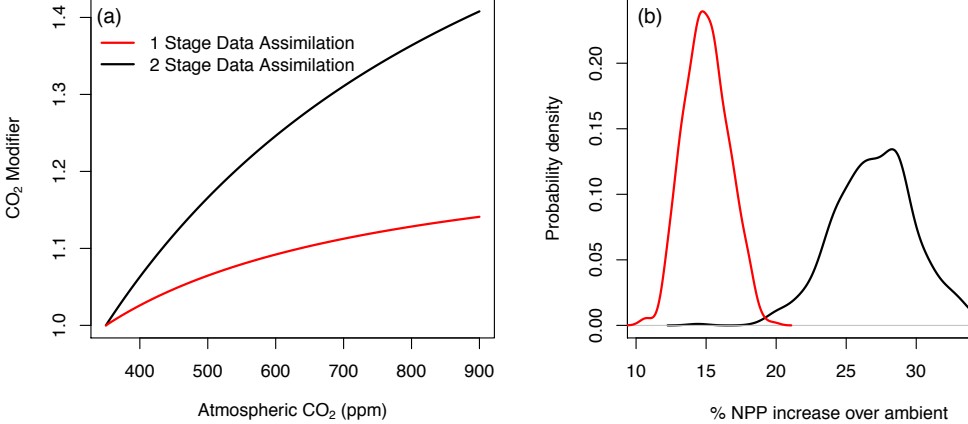

Figure 5. The influence of the data assimilation approach on predictions of how productivity
responds to atmospheric $CO_2$. (a) The relationship between atmospheric $CO_2$ concentration and
the modifier of light-use efficiency when all plots and experiments are assimilated
simultaneously (1 stage) and when the Duke and NC2 plots are assimilated before assimilating
the remaining observations across the region (2 stage). (b) The probability distribution of
predicted response of NPP to the elevated $CO_2$ at the Duke FACE experiment for the two
assimilation approaches. Uncertainty was estimated by integrating the parameter uncertainty
estimated through data assimilation (see Methods).

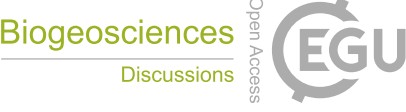



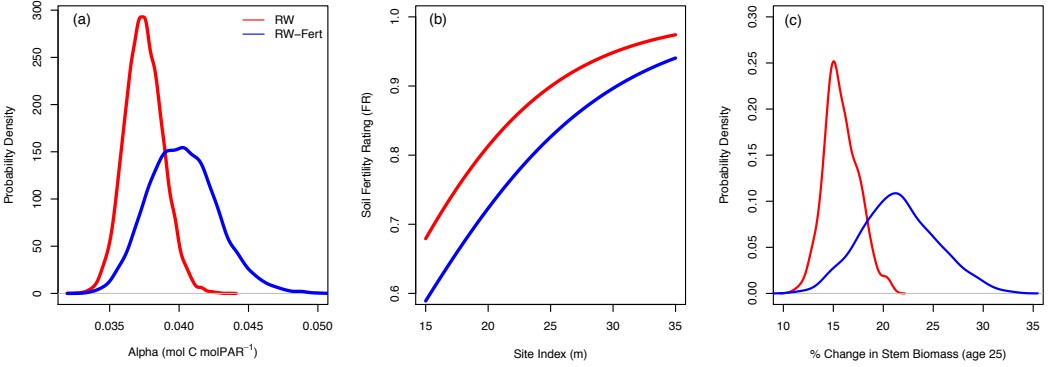

Figure 6. The influence of the data assimilation (DA) approach on predictions of light-use
efficiency, nutrient limitation, and how productivity responds to nutrient addition. (a) The
posterior distribution for the potential light-use efficiency parameter (alpha). (b) The relationship
between site index and the nutrient limitation modifier in 3-PG (FR). (c) The predicted
distribution of the response of stem biomass to nutrient fertilization (setting FR = 1) at the focal
site in Georgia. The red line corresponds to DA that included nutrient addition experiments. The
blue line corresponds to DA that did not include nutrient addition experiments.




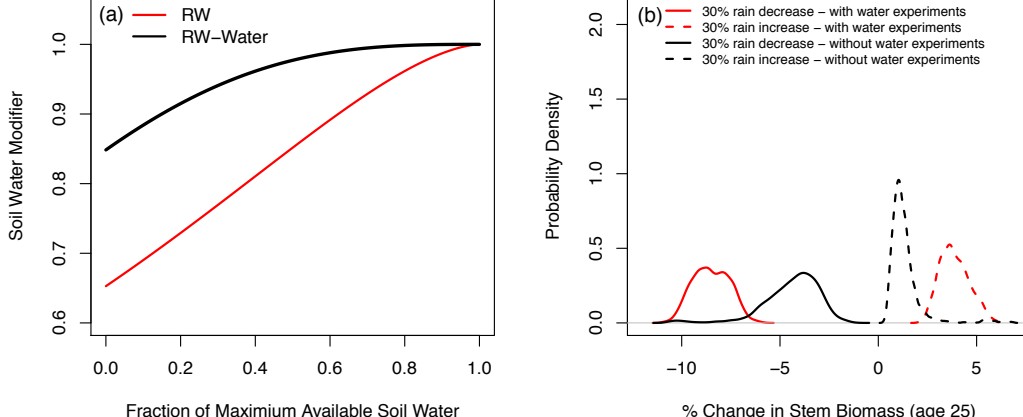

Figure 7. The influence of the data assimilation (DA) approach on predictions of water limitation
and how productivity responds to a change in precipitation. (a) The relationship between fraction
of maximum available soil water, as predicted by 3-PG, and the modifier of light-use efficiency
and canopy conductance. (b) The predicted distribution of the response of stem biomass to a 30%
increase (dashed lines) and a 30% decrease (solid lines) in precipitation at the focal site in
Georgia. The red line corresponds to DA that included water manipulation experiments. The
black line corresponds to DA that did not include water manipulation experiments.






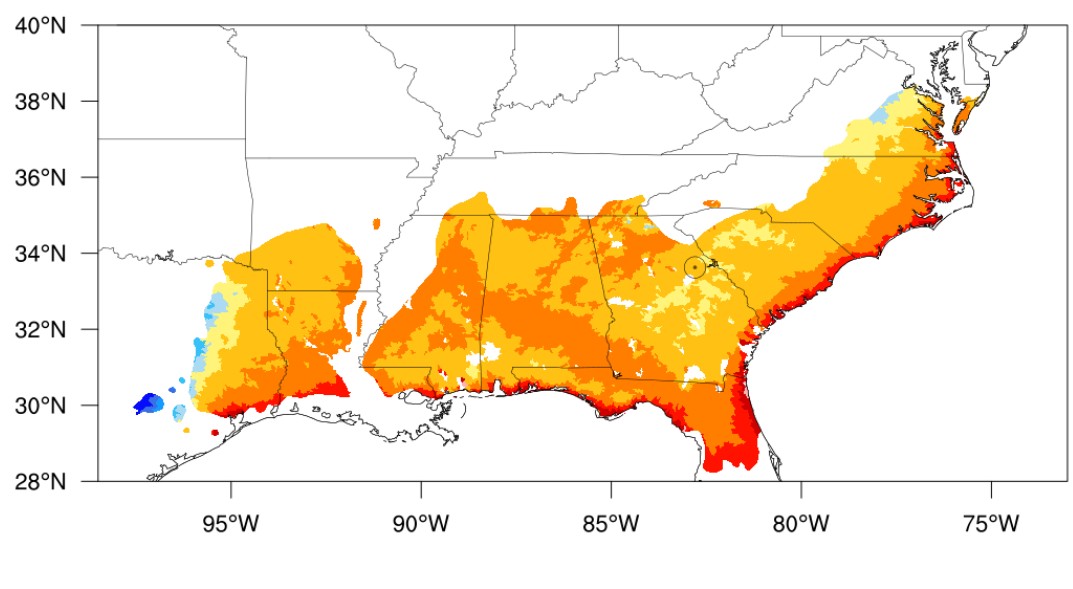

Figure 8. Regional predictions of stem biomass stocks for a 25-year-old stand planted in 1979.
Parameters used the predictions were from the RW data assimilation approach described in Table
5. The focal site in Georgia highlighted in Figures 5c and 6b is represented by the circle
containing the dot.






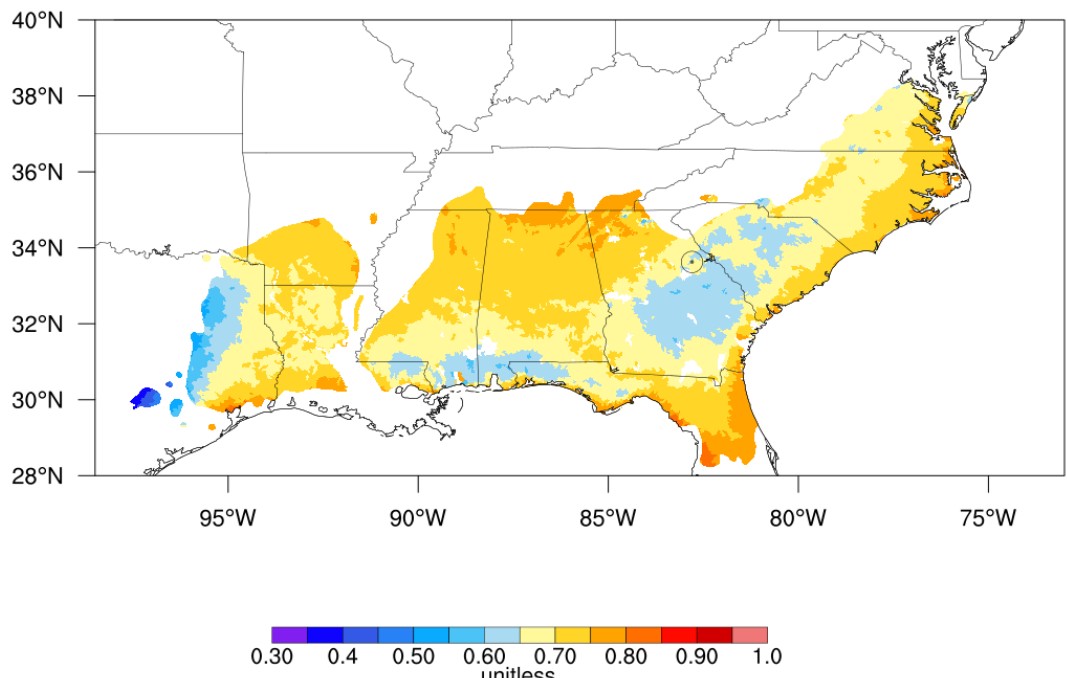

Figure 9: Regional predictions of the soil fertility factor (FR) used in 3-PG to define the nutrient
status of the simulated stand. Parameters used the predictions were from the RW data
assimilation approach described in Table 5. The focal site in Georgia highlighted in Figures 5c
and 6b is represented by the circle containing the dot.






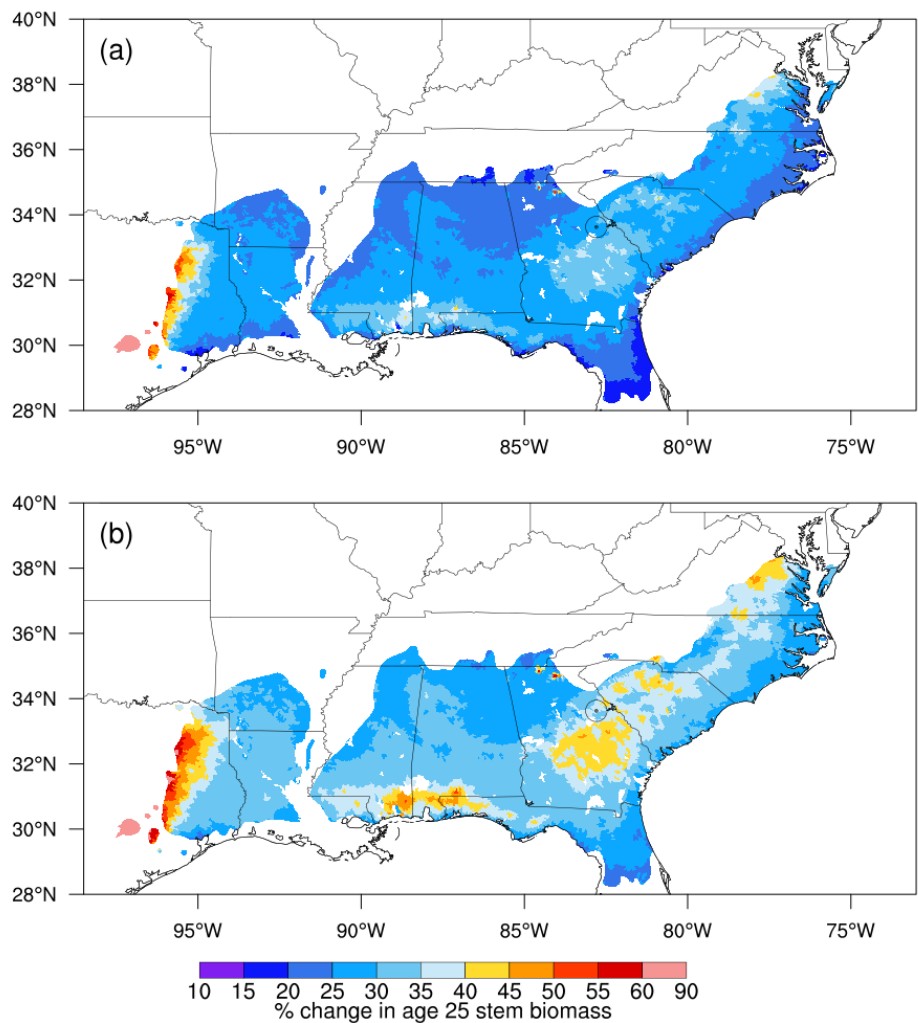

Figure 10: Regional predictions of the change in stem biomass of a 25-year stand when nutrient
limitation is completely removed through nutrient addition (simulated by setting FR = 1).
Predictions from data assimilation that included nutrient addition experiments are shown in (a)
and prediction data assimilation that did not include nutrient addition experiments are shown in
(b). The focal site in Georgia highlighted in Figures 5c and 6b is represented by the circle
containing the dot.



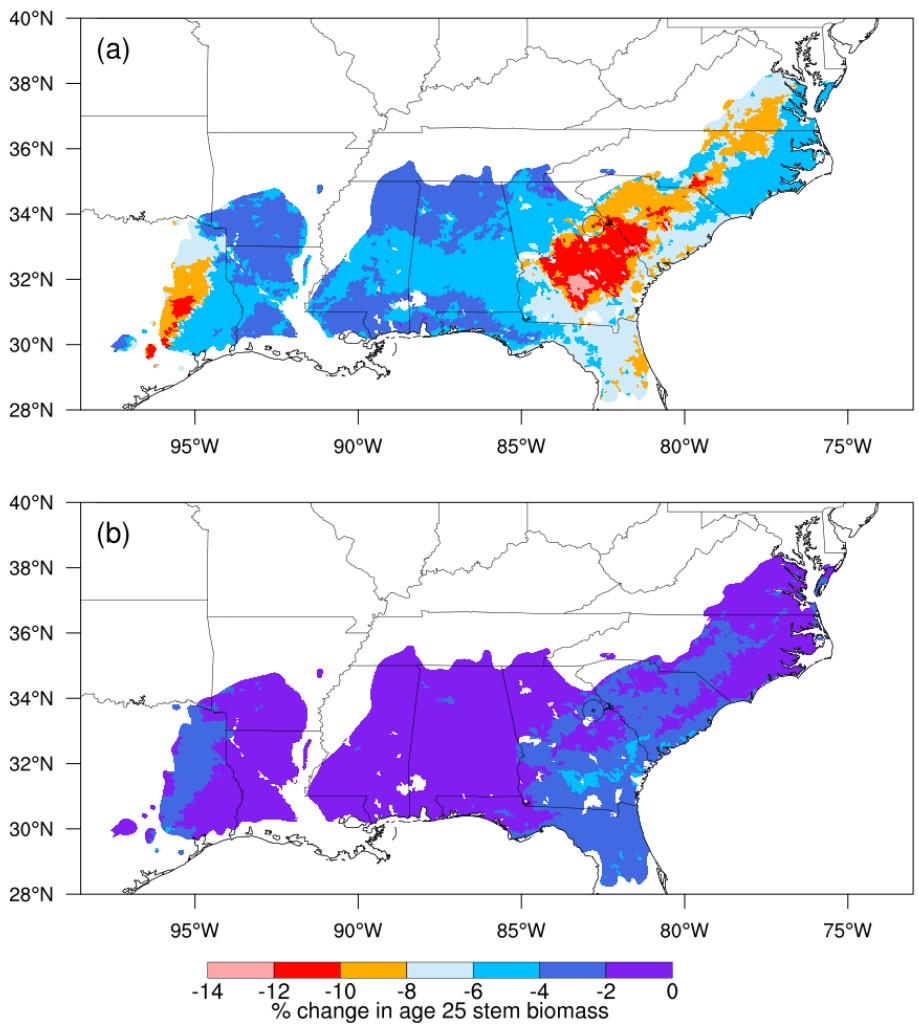

Figure 11. Regional predictions of the change in stem biomass of a 25-year stand when annual
precipitation is reduced by 30%. Predictions from data assimilation that included water
manipulation experiments are shown in (a) and prediction data assimilation that did not include
water manipulation experiments are shown in (b). The focal site in Georgia highlighted in
Figures 6c and 7b is represented by the circle containing the dot.