# Peer review of "Leveraging 35 years of *Pinus taeda* research in the southeastern U.S. to constrain"

_Biogeosciences, 2017_

## Referee Comment (RC1) · AP Walker (Referee) · 23 Mar 2017

Thomas et al present a data-assimilation (DA) study using constraints from multiple data streams from multiple sites and experiments to optimise parameters in the monthly timestep PG-3 model of loblolly pine production. The study has three specific objectives. Stated on lns 170-171, 1) a new regional and hierarchical data assimilation system with the capacity to assimilate multiple data streams from multiple experiments; stated on ln 179-180, 2) the consequences for parameter estimation and prediction of including or not including ecosystem manipulation experiments (this could be more

broadly stated as evaluation of the DA); and stated on ln 181 3) model predictions with the optimised parameter set of forest biomass changes in response to changes in nutrient addition of precipitation. This study is well thought out and implemented, presents a useful advance to the use of DA in ecosystem modelling and forecasting, and will likely be of interest to many readers of Biogeosciences. My main criticism is that the distinction between the three areas of this study is often not made explicitly throughout the manuscript and consequently the manuscript is not as readable or as clear as it could be. The majority of my comments are an attempt to help improve the organisation and presentation of the manuscript with the goal that this study will be as widely read and cited as possible.

With that in mind, I suggest organising the manuscript as much as possible by the three stated objectives. I suggest combining the sentence on lns 179-180 with the sentence on lns 170-171 and explicitly listing the three objectives together. The results and discussion section would benefit from organisation along the lines of the three stated objectives. I suggest breaking each into three subsections, each dealing with one of the objectives. Again the conclusions section should specifically address each objective.

Abstract

It would be good to be specific about who the target audience is for this research. The research straddles a technical field that develops DA but the technique produces a tool at a level of maturity that could be used by foresters. These ultimate end users could be more explicitly targeted.

Introduction

Is a bit long and could a page or so could be cut without loss of content. Paragraphs on lns 82-105 could be combined and reduced in length. The main point is that ecosystem experiments can help to reduce the problem of equifinality in DA. The paragraph on lns 108-141 makes some nice points but could be substantially shortened without loss of

content. Much of the paragraph is methods like.

Weight to rare experiments (mentioned on ln 125) could also apply to rare data types. Later in the paragraph (ln 135-136) the authors state that data of different frequency is a problem in biasing the cost function toward high frequency data, but offer no solution other than a monthly timestep model. Rare data, or low frequency data, could also be given higher weights. Also high frequency data could be summarised at lower frequency.

Methods

Again long and could probably be made more concise. Also the organisation is tough to follow. I suggest leading with the observations, the various sites, and measurement campaigns/projects. Many of these are not properly introduced. This will provide a comprehensive introduction to the system and what measurements actually go into this DA system. Observation sites and projects are mentioned on ln 409-410, but these are not introduced and need to be described in the observations section of the methods.

I found section 2.3 very difficult to follow. I'm not expert on DA mathematical methods but I have a reasonable conceptual handle on DA, and yet I was lost in the first paragraph. I also ran this section by a colleague who is expert in the mathematics underpinning DA and they agreed that this sections needs to be clearer. Their key criticism was that they could not see the derivation of Eq 7, perhaps the authors could add the derivation to an appendix. And that it is not clear how the MCMC was used to sample Eq 7. A clear description of the details of the MCMC procedure is necessary, along with the presentation of the cost function. Also the first term on the righthand side of Eq 7 is not the same as the righthand side of Eq 1, is this deliberate? And E is never defined.

I strongly suggest reworking section 2.3 of the methods to be extremely clear about the DA process and how it was implemented. Start with a clear description of the goals of the DA – state estimation and estimation of parameter distributions. Then describe all

the various sources of uncertainty and how the method accounts for them. Then take the reader step by step through the method. Perhaps a diagram would be useful. The following comments are an attempt to provide examples of where confusion arises but they are in no way comprehensive. The sentence on lns 281-283 is more or less stating the the the same thing as the sentence on lns 284-285. I suggest fusing these together. Is the reference to a "latent model" really necessary, it is confusing with the mathematical model. Would "true" system states and fluxes convey the same meaning? Do not try to justify the method in comparison with previous methods (e.g. lns 286-291), in the methods this just confuses the description and this can be argued in the discussion. On lns 291-293, this is state estimation right? That's fine but is it really the focus of your method? None of the three stated objectives are for state estimation. How exactly was estimation of the latent state or flux the first step in the process when it includes the optimised parameters etc as described on lns 296-298? Seems like the statement on ln 306-308 should come before the previous paragraph.

Section 2.4 jumps around between objectives. Some text would fit better in section 2.3, for example lns 408-428. Text on lns 454-461 would be better organised if it were to follow the text on 430-444, then the regional simulations can be presented afterwards.

I suggest defining sections 2.3, 2.4, and an additional 2.5 to be organised by the three stated objectives.

Also, while commonly used by the modelling community, I do not agree that you can run "experiments" with models. Models make predictions from a specific set of mathematical hypotheses and defined scenarios. An experiment is designed to test predictions and discriminate among hypotheses.

Results

Why were only 31 parameters optimised, can you describe why this set were chosen from the total 46?

Technically the parameters are not "sensitive" (ln 480), it is the model output that is sensitive to the parameter. "Influential" would be a better adjective to describe the parameters.

Lns 486 & 488 variability is described as being reduced but no data are provided. Can you quantify these statements. There are many statements like this throughout the results and they ought to be quantified (e.g. lns 502, 508). Also on 508, is mean correct, isn't this the median of the parameter distribution?

Some kind of visual representation of the data in table 5 would be useful.

Ln 492 what do you mean strong priors? Well defined from measurements and literature with low variance? Could you quantify this?

Lns 494 the process uncertainty parameters are mentioned here and in the methods, but results are barely presented (only in the supplement) and are not discussed, or not that I noticed. This is a very interesting concept and I would like to see these data presented a little more and at least a little discussed. What kind of impact does including these parameters have on the optimised parameter distributions? I understand you are already presenting a lot, but this is fairly novel as far as I'm aware and is of interest.

Figure 10 and 11 would be more in keeping with your stated goal of forecasting on lns 65-68 if you removed the b panels in both plots. If you think that the parameter estimates when including the data from the manipulations gives a better estimate of those parameters then the data in panels b are not particularly useful for forecasts. In my view, and as stated on line 67 & 68 "provide information on both the expected future state of the forest and the probability distribution of those future states", the final figures would be much stronger if the probability distribution of the future states shown on the a panels were represented on the b panels.

While it is interesting to show the consequences for prediction of inclusion of manipulations or not, and the opposite sign of the change in predictions when water and nutrient

manipulations are included, you already show this in Figures 6 & 7. If you want to keep the b panels in 10 and 11 I suggest you add them as extra panels to figures 6 & 7, showing the absolute delta (or similar) from the simulations that include the manipulation delta. This will allow you to address the question: what are the consequences of not including data from manipulations? Without confounding the predictions from the most appropriate DA product for the scenarios tested. Also the scale ought to be the same for the data presented in Figs 10 and 11.

Was $CO_2$ change included in the above projections of removal of nutrient limitation and precipitation reduction? Furthermore, it seems you have included data from water manipulation experiments, nutrient manipulation experiments, and $CO_2$ manipulation experiments. But you have only made projections for nutrient and precipitation change. Why not $CO_2$ change? $CO_2$ projections would complete the study.

Additional points

I think the title would benefit from the addition of "Loblolly Pine".

Ln 50 Duke FACE experiment had 4 replicate plots, so where does the 5 come from on this line. An additional plot from the unreplicated prototype?

Ln 48 – 50 the sentence on this line would help flow if it were before the preceding sentence.

Ln 65 I don't think I would classify the three areas mentioned in the previous sentence as tools. They are more than tools, they are also knowledge.

Ln 67 What do you mean by "based on" here. Can probably delete. Also while I think your methods could be used for "forecasting" you don't really use the method in that sense.

Ln 73 insert "can" in between "that generate"

Ln 85 86 "carbon allocation and turnover" This is worded a little awkwardly

Ln 97-99 awkward way to start a paragraph.

Ln 111 suggest replacing "important" with "useful" or something more descriptive

Ln 155-157 suggest replacing "nutrients" with "nutrient addition". Also suggest removing hyphens.

Ln 162-163 Awkward

Ln 171 Again I think you need to call out loblolly pine here

Ln 175 The authors chosen acronym, in my view, somewhat undersells what they are doing. The DA method is hierarchical and considers data from multiple sites and of multiple different types. The acronym gives not indication of this and suggests that the DA method is only suitable for Pine Plantations. Of course it is the authors' choice though.

Ln 307 insert "considered" between "was a"

Ln 446 replace "regional" with "region"

Ln 522-524 I'm not sure what you mean here, could you clarify?

Ln 528 delete "a"

Ln 576 replace "detangling" with "disentangling"

Ln 582 I think "synthesised" would be a better word to use than "organised"

Ln 591-591 I take your point about equifinality but can you really say this if predictions were not improved in some way? Just a thought. Is there a way that you can be sure that the mechanisms were correctly distinguished?

Ln 633-634 Agreed, but did your method strictly weight the data? Wasn't it more that the hierarchical method gave priority to the $CO_2$ manipulation data?

Ln 646 replace "than" with "that"

Ln 656 quantify this statement

Ln 662-663 this was news to me when I read this sentence. I think this would become clearer once the methods can be clarified as suggested above.

Ln 668 suggest changing "prior" to "previous", just to maintain the meaning of prior in the Bayesian sense.

Ln 673 you do not show any data on covariation of parameters.

Ln 676-680 I like this statement, makes a lot of sense. But is it most appropriate here? This point should be made clearly in the methods.

Ln 685 suggest deleting "Multivariate Constructed Analogs (MACA)" it is not needed.

Ln 692-697 This is a good point but I'm curious why the change in biomass in response to precipitation reduction was small given the large change in parameter values when water manipulations were included in the DA. Can you try to explain this based on the process hypotheses embedded in the model.

Ln 698 replace "reduced" with "reduction"

Ln 707 insert "as a function of"

Ln 719 insert space in "fromadditional"

Ln 760 While I'm sure the methods and tools developed by this study could be used for ecological forecasting, strictly speaking this study is not ecological forecasting. The third objective, which concerns optimised model predictions, is a scenario analysis rather than a forecast.

Ln 769 no need to cite Medlyn et al 2015 here

---

## Referee Comment (RC2) · Anonymous Referee #2 · 11 Apr 2017

Quinn Thomas et al. present a model-data fusion, or data assimilation, study that gathers 35 years of carbon cycle-related observations and manipulation experiments taken in Loblolly Pine ecosystems in the Southeastern US to optimize parameters of the 3-PG model within their new framework DAPPER. The authors examine the ability of the observations to constrain model parameters using a number of approaches for assimilating the different types of data, and they further examine the differences in model behavior/sensitivity and change in biomass stocks across the southeastern US as a result of the different experiments.

[Figure]

The authors have carried out an impressive and exhaustive collection of data for constraining the 3-PG model in this study. This, and their investigation into different approaches for assimilating different types of data, in particular manipulation experimental data, make this study a noteworthy contribution to model–data assimilation literature in forested ecosystems, and therefore I would recommend publication in Biogeosciences. However, as it stands the manuscript is quite long and dense, which is understandable given the amount of detail that is required to present such a wide array of data and experiments. This being said, I recommend that the authors try to edit the article following some of the suggestions below (and their own views) to improve the clarity and readability of the text before this article is published.

Overall, the objectives and key points of this study can get lost in the text. I think a few more sub-sections in the main text and supplementary, references and links between sections would help the reader to better follow and absorb the necessary amount of detail presented in the manuscript. I would also find it useful if the authors posed a few key scientific questions to help them highlight the main messages of the study.

Some sections in the methods could do with more explanation for why certain approaches were used (see comments below) or better links to the supplementary material, as I have just mentioned. The introduction and discussion are quite long and this can prevent some of the key points from being highlighted. I suggest the authors try to cut down the text where they see fit, including some sentences that essentially are repetitions of earlier statements.

The paragraphs in the results section could be separate sections with sub-headings in order to guide the reader, while at the same time the results could benefit from stronger links between each section, especially before line 522, in particular comparing the between the 1st and 2nd stages, or the different 2-stage approaches with the 1-stage approach. At the moment, the results section before line 522 is a bit fragmented, making it harder to weave together a coherent story that brings out the key points.

Reading this manuscript I found myself asking: What do you expect from each experiment/approach? What will you gain/lose? Which approach is the right approach, going forward? These questions were largely answered in the discussion, and therefore I have made a suggestion below that perhaps some of the results and discussion could be merged within the sub-sections suggested above. This is a personal style issue however.

Finally, the authors may consider cutting other sections of the discussion that are not fully pertinent to the results as the paper is already quite full of detail. I would like to stress that despite this suggestion I did find the discussion to be interesting and comprehensive, but I would like to see the key messages highlighted more and am concerned the length of the paper may overwhelm the reader.

Introduction

Line 97: "relative contribution of each environmental control should be separated in order to correctly parameterize the sensitivity to changes in the environment". I agree to some extent but this is very hard to do and should we be separating each environmental control, as the interaction between different environmental changes may produce different outcomes than if each were treated separately? I would be interested to hear the authors thoughts on this and what they think the impact of assimilating manipulation experiments data separately has on their results.

Line 124-128: See previous studies Wutzler and Carvalhais (2014) and Section 2 of MacBean et al. (2016) for further discussion on debate of how to deal with the issue of weighting to account for the number of observations and/or using a multi-stage assimilation approach to address challenges of assimilating a diverse set of observations. Both issues are the subject of debate in the literature. On the issue of weighting by the number of observations, from a mathematical standpoint there would be no need if the error covariance matrix is properly characterized; however, this is difficult to achieve in practice. Similarly, a joint or simultaneous assimilation, in which all observations are

assimilated together, is mathematically more rigorous as the error covariance between the observations can be properly taken into account. I appreciate that you have discussed the benefit of weighting by the type of data in the discussion, but this debate in the literature (for and against weighting, due to the abovementioned reasons) should perhaps be referred to more clearly in this study.

Line 129: It is true of course that to constrain changes in biomass monthly time-scale models are sufficient, but note that monthly time-scale models are not the only way to overcome computational challenges associated with inverting a complex ecosystem model. There are sophisticated yet simple algorithms that dramatically improve the sampling of parameter space in a limited number of iterations. See the work of Jasper Vrugt: https://scholar.google.co.uk/citations?user=zkNXecUAAAAJ&hl=en&oi=ao

Methods

Section 2.1 It would be good if you could refer to references and/or relevant sections in the Supplement in Section2.1 to depict between standard characteristics of the 3PG model specific additions or alternative choices you made and (and to explain why you made those choices). For example: - Line 201-202: Was this additional function based on a published study? - Line 209: Is the site-index a new addition to the model that you developed? If so, from where? - Lines 218-220: Why did you remove the dependence of total root allocation on FR for the DA study? - Line 229-231: A reference for or further explanation of this modification would be good here. - Line 245: "implicit irrigation in very dry conditions." Is this a realistic feature of these sites? How does this affect the results? Especially for the water availability manipulation experiments.

Line 250: do you mean to say "mean monthly GPP"?

Line 251-252: How did you select the 31 parameters to be optimized?

Table 1: Please can you give the equation for how the sensitivity is calculated? Also, please could you explain why there is both a number and "vague" given for the uncertainty of some parameters? If "vague", please can you detail how you defined the prior uncertainty/ranges in the text? Finally, I appreciate you have a lot of information to convey and the tables are large, but it might be good to have all optimized parameters here and just indicated which ones are referred to in the discussion. As a general comment, it is hard to find some of the information you refer to in the Supplement (e.g. the other optimized parameters you refer to in the caption of Table 1). Please could you split the Supplement into numbered/indexed sections and then refer specifically to the relevant section to help the reader?

Line 255-265: How did you initiate the biomass pools? Based on site-level data for the start of the simulation period? Please detail with references. If no site data were available, how sensitive were your DA experiments on the method used to initiate the biomass pools? Later note: I see you have addressed this in Section 2.4. It might be useful to refer to that section here so the reader is not questioning this in this section.

Section 2.2 Table 2: Last column – Table 3 instead of Table 4. Also, please could uou explain, or give references, for why the SD for observations sometimes varied between 10% and 2.5% of the observation.

Section 2.3 Equation 4: Please explain why you picked a uniform distribution between 0.001 and 100? Lines 348-349: Please explain why (only) 3 MCMC chains were run? Was a convergence metric such as R-hat used?

Section 2.4 Lines 398-399: Although I understand the reasoning that these sites are close together and the most data rich, I don't understand why you lump the Duke CO2 enrichment site with DK3 and NC2 in the 1st stage when you stated that you wanted to test the influence of the CO2 fertilization – why not just test the Duke CO2 enrichment site by itself in the 1st stage and the remaining sites/plots in the 2nd stage to answer this question?

Further to the above point, I appreciate the extra experiments to understand the influence of the CO2 fertilization on the posterior parameters, and the further experiments

to determine the influence of the water treatments and nutrient addition. But how dependent are your results on which type of observation and/or treatment is assimilated in the 1st stage vs 2nd stage? Would the results different if you reverse the stages you have in your current set-up? Again, see Wutzler and Carvalhais (2014) and/or MacBean et al. (2016) who discuss these issues (as well as the issue of the weight of different types of data, as you discuss below. A pseudo-test with synthetic observations would have been useful prior to assimilating real data to determine whether the exact set-up of a 2-stage assimilation is sensitive to the order of observation assimilation as well as to confirm if the assimilation system is able to constrain the parameters to their correct values.

Lines 430-465: While the tests and approaches put forward here are interesting, the text is dense. Any efforts the authors could make to simplify the description of the experiments and simulations performed (perhaps with the use of a table and simulation/experiment code names?) would likely help the reader.

Lines 467-475: The cross-validation exercise presented here is a useful one. Was a similar test used to assess the validity of the posterior distributions of the manipulation experiments, even though there are fewer sites?

Results

Line 480-484: Description of the sensitivity analysis and choice of parameters should be in the methods. Was this a one-at-a-time sensitivity analysis or a full global method? What is the justification for using this approach versus an existing global sensitivity analysis that accounts for correlations between parameters and explores the whole parameter space (unless I have misunderstood what was done)? Why did you fix the light extinction coefficient as opposed to the quantum yield parameter?

Supplemental Table 3 and Table 5: As mentioned above I would suggest having all the optimized parameters in one table. I would also suggest putting the prior min/max in Table 5 even though it might mean having an extra line/column per parameter and

taking this information out of table 1 so it is easier to see how well the optimization has constrained the parameters. Finally, I would suggest splitting up the parameter tables into the sections you refer to in the text, e.g. "temperature sensitivity of quantum yield" or "physiological parameters" etc. This will make it easier for the reader to refer to the tables when reading the text.

Which experiment do the supplemental figures correspond to? The "ALL" experiment? This should be detailed.

Are you talking about the 1st stage experiment in the first paragraph of the results? If so, it would be good to specify this, and I would further suggest splitting the results into sections to more easily guide the reader.

Do you discuss DK+NC2-fert in the results, or have I missed it? Perhaps more needed on the 1-stage versus 1st and 2nd stages before you discuss the experiments with and without nutrient and water addition (i.e. before line 522)?

Figure 5 comes before Figure 4 in the text – switch around?

Lines 507-515: I am a bit confused by the sentence "The two-stage assimilation was critical for constraining the CO2 quantum yield enhancement parameter (Calpha700)" as you then go on to say (and show, in Figure 5) that the 1 stage resulted in a narrower uncertainty interval? I guess you mean that despite the higher 95% confidence interval, the 2-stage approach results in a more realistic parameter value but I am not at all sure on that? Please could you clarify this in the text?

Line 517: I would suggest putting the names of the soil fertility parameters in brackets to aid the reader, or again put sub-headings in the parameter tables.

As you did not have a strong difference in predictive capability between experiments with and without nutrient or water addition, even though you had different parameters, that presumably means you have a certain amount of model equifinality? You discuss and show the difference in model behavior as a result of the different approaches in

Figures 5 – 7, but you do not discuss which one you think leads to the right behavior? Do you have an idea? Perhaps a synthetic experiment with pseudo-observations taken from the model simulations might help with this (a so-called "observing system simulation experiment", or OSSE)?

Lines 522 onwards show very interesting results. However, I would suggest that the patterns detailed in last two paragraphs (Lines 553-572) would benefit from explanations linking back a bit more (not just referring to figures) to the different model behavior/mechanisms identified and discussed in the RW-fert and RW-water sections just above.

Discussion

First paragraph is more of a summary than a discussion and could be cut or added to conclusions.

Although perhaps a little too long, this is a useful discussion that ties the results together and answers some of the questions I raised in my comments on the results. Perhaps it would be useful to combine some of the summary points raised in the discussion with relevant sections in the results with separate sub-headings as I mentioned above.

Lines 650-652: Interesting point and in addition, as I have mentioned above, I think a synthetic experiment would also be very helpful in this regard.

Minor comments

Line 87: Do you mean the "assimilation of manipulation experimental data", rather than the "assimilation of experiments"?

Line 88: two or more

MacBean, N., Peylin, P., Chevallier, F., Scholze, M., and Schürmann, G.: Consistent assimilation of multiple data streams in a carbon cycle data assimilation system, Geosci.

Model Dev., 9, 3569-3588, doi:10.5194/gmd-9-3569-2016, 2016

---

## Author Response (AR1)

Overall response to both reviewers

We greatly thank both reviewers for the thorough and very helpful reviews. Synthesizing the two reviews indicated that the manuscript had a lot of interesting information but was too dense to effectively communicate the key ideas. In response, we have simplified the analysis so that it has fewer moving parts. Our reanalysis also represents improvements to the data assimilation approach that have occurred since the manuscript was first submitted.

We simplified and modified the analysis as follows:
1) We removed the need for the 2-stage data assimilation. Now there are two chains that assimilate all sites simultaneously: one that includes site-specific parameters for only the Duke site and one that does not include the site-specific parameters. This modification allows the analysis to focus on why the parameters are different rather than focusing on the need to weight the Duke site differently. Since we did not actually weigh the Duke site differently in the original analysis nor include a synthetic experiment that explores the influence of site weighing on parameter inference, we feel that the simplified, updated approach is more sound and easier to understand.

   *The previous text on the two-stage vs. one-stage assimilation is now condensed to the following:*

   *In Methods:*

   *During preliminary analysis, we found that the Base assimilation predicted lower stem biomass than observed in the elevated CO2 plots in the Duke FACE study. Further analysis investigating the cause of the bias in the CO2 plots showed that three parameters (wSx1000, ThinPower, and pCRS) were required to be unique to the Duke FACE study in order to reduce the bias. Therefore, the Base assimilation included unique parameters for wSx1000, ThinPower, and pCRS parameters in all plots in the Duke FACE and US-DK3 studies. To highlight the need for the site-specific parameters, we repeated the Base assimilation approach without the three additional parameters for the Duke studies (NoDkPars assimilation).*

   *In Results:*

   *The plots at the Duke Forest study had a higher carrying capacity of stem biomass before self-thinning (WSx1000), smaller self-thinning parameter (ThinPower), and lower allocation to coarse root (pCRS) than values optimized from the other plots across the region (Table 6). The DA approach without these three study specific parameters (NoDkPars) predicted significantly lower accumulation of stem biomass in response to elevated CO2 than observed (df = 4, p = 0.002; Figure 5). The NoDKPars assimilation optimized the CO2 fertilization parameter (fCalpha700) to a value that predicted 45% less light-use efficiency at 700 ppm (1.13 in NoDKPar vs. 1.33 in Base; Table 6) than the Base assimilation.*

*In Conclusions:*

*Constraining the sensitivity to atmospheric $CO_2$ differs from constraining the sensitivity to ASW because, unlike the multiple constraints on water sensitivity (drought, irrigation, and gradient studies), environmental conditions created by the few elevated $CO_2$ plots provided unique constraint on parameters. Our finding demonstrated that DA efforts should test for bias in unique ecosystem experiments before finalizing a set of model parameters used in optimization. In particular, we found that the parameter governing the photosynthetic response to elevated $CO_2$ (fCalpha700) was substantially lower when all parameters were assumed to be shared across all plots than when the $CO_2$ fertilization experiment was allowed to have unique parameters. The need for the three unique parameters at the Duke FACE study parameters can be explained by the constraint provided by multiple data streams and multiple plots. An assumption of the model was that an increase in stem biomass caused a decrease stem density through self-thinning, unless the average tree stem biomass was below a parameterized threshold (WSx1000). Therefore, an increase in photosynthesis and stem biomass through $CO_2$ fertilization could cause a decrease in stem density. For a single study, it is straightforward to simultaneously fit the $CO_2$ fertilization and self-thinning parameters to fit stem biomass and stem density observations for the site. However, regional DA presents a challenge because the self-thinning parameters are well constrained by the stem biomass and stem density observations across the region but the $CO_2$ fertilization parameters are not. As a result of the regional DA, the self-thinning parameters caused a stronger decrease in stem density than observed in the Duke FACE study. Therefore, the optimization favored a solution where there was a lower response to $CO_2$, thus a smaller decrease in stem density. Allowing the Duke FACE study to have unique self-thinning parameters that resulted in lower rates of self-thinning and allowed for simulated stem biomass to respond to $CO_2$ in a way that matched the observations without penalizing the optimization by degrading the fit to the stem density.*

*Our finding that the Duke FACE study required unique self-thinning parameters to reduce bias in the simulated stem biomass suggests that when using DA to optimize parameters that are shared across plots, careful examination of prediction bias in key sites that provide unique constraint on certain parameters (like the Duke FACE) is critical. Based on this example, we suggest that DA efforts using multiple studies and multiple experiment types identify whether particular experiments at limited number of sites have the potential to uniquely constrain specific parameters. In this case, additional weight or site-specific parameters may be needed to avoid having the signal of the unique experiment overwhelmed by the large amount of data from the other sites and experiments. Additionally, the finding suggests that multi-site DA should consider using hierarchical approaches to predicting mortality, particularly because mortality is often not simulated as mechanistically as growth. A hierarchical approach, where each plot has a set of mortality parameters that are drawn from a regional distribution, could avoid having unexplained variation in mortality rates lead to bias in the parameterization of growth related processes (i.e., growth responses to $CO_2$, drought, nutrient fertilization, etc.). The hierarchical approach to mortality could also highlight patterns in mortality*

*rates across a region and allow for additional investigations in the mechanisms driving the patterns.*

2) We replaced the assimilations that separately removed the water and nutrient experiments with a single assimilation that removes all experiments (water, nutrient, and $CO_2$). We feel this is a better approach because the analysis included multi-factor experiments. For example, in the previous analysis, the removal of nutrient experiments also removed $CO_2$ and drought treatments. Now we present two sets of optimized parameters: with and without experiments. This allows us to more clearly address the question "how do the parameter distributions depend on the inclusion of ecosystem experiments in the data assimilation". Some of the figures were simplified in the process of this revision.

*The previous text because the assimilation with and without experiment is now condensed to the following:*

*Methods:*

*We also evaluated how parameter distributions and the associated environmental sensitivity of model predictions depended on the inclusion of ecosystem experiments in data assimilation. First, we repeated the Base assimilation, this time excluding the plots that included the manipulated treatments (NoExp). We removed all manipulation types at once, rather than individual experimental types, because all experimental types were involved multi-factor studies. The NoExp assimilation had the same number of data streams as the Base assimilation because it included the control treatments from the experimental studies. The NoExp assimilation represented the situation where only observations across environmental gradients were available. Second, we compared the parameterization of the ASW, soil fertility, and atmospheric CO2 environmental modifiers from the Base to the NoExp assimilation. The modifiers equations are described in Supplemental Material Section 1.2 and 1.3. Third, we repeated the same independent validation exercise for the 160 FMRC plots as described above for the Base assimilation. Fourth, we predicted the treatment plots in the irrigated, drought, nutrient addition (only plots where FR was assumed to be 1), and elevated CO2 plots. As for the Base assimilation, we used a t-test to compare the experimental response between the NoExp assimilation and observed and between the NoExp and Base assimilations. Since the experimental treatments were not used in the optimization, this was an independent evaluation of predictive capacity.*

*Results:*

*Excluding the experimental treatments from the data assimilation did not strongly influence the predictive capacity of the model. The RMSE validation plots in NoExp assimilation decreased slightly compared to Base assimilation (21.8 to 18.0 Mg ha-1) while the bias slightly increased (-3.7 to -4.1%)(Figure 4b). Excluding the experimental treatments resulted in a significantly lower response of stem biomass to elevated CO2 than observed (df = 4, p < 0.001; Figure 5). Furthermore, there was a slight negative response of stem biomass to CO2 in the NoExp assimilation because the parameter*

*governing the change in foliage allocation at elevated CO2 (fCpFS700) was unconstrained by observations (Table 6). This led to convergence on the lower bound of the prior distribution (0.5) where foliage allocation decreased with increased atmospheric CO2. The predictions of irrigation, drought, and nutrient addition experiments were not significantly different between the Base and NoExp assimilations (Figure 5).*

*The parameters and associated response functions in the 3-PG for nutrients, ASW, and atmospheric CO2 differed between the Base and NoExp assimilations (Figure 6). First, the parameterization of the soil fertility rating (FR) showed a stronger dependence on SI in the NoExp assimilation than in the Base assimilation (Figure 6a). For a given SI there was a lower FR, thus stronger nutrient limitation, when experimental treatments were excluded from assimilation. Second, the parameterization of the function relating photosynthesis and canopy conductance to ASW resulted in lower photosynthesis and maximum conductance when soil available water was less than 50% in the NoExp than Base assimilations (Figure 6b). Finally, the response of photosynthesis to atmospheric CO2 was functionally zero in the NoExp assimilation, thus highlighting the importance of the elevated CO2 treatments in the Duke FACE study for constraining the parameterization of the CO2 response function (Figure 6c).*

*Discussion:*

*The most important experimental manipulation for constraining model parameters was the Duke FACE CO2 fertilization study because the CO2 fertilization parameters (fCalpha700 and fCpFS700) converged on the lower bounds of their prior distributions when the experiments were excluded from the assimilation. In contrast, excluding the nutrient fertilization, drought, and irrigation studies did not substantially alter the predictive capacity of the model. This finding suggests that data assimilation using plots across environmental gradients alone can constrain parameters associated with water and nutrient sensitivity. However, regardless of whether the experiments were included in the assimilation, the optimized model predicted higher sensitivity to drought than observed, highlighting that future studies should focus on improving the sensitivity to drought.*

*The 3-PG model included a highly-simplified representation of interactions between the water and carbon cycles that resulted in parameterizations that may contain assumptions that require additional investigation. First, transpiration was modeled as a function of a potential canopy transpiration that occurred if leaf area was not limiting transpiration. The LAI at which leaf area was no longer limiting was a parameter that was optimized (LAIgcx in Table 5), resulting in a value of 2.2. Interestingly, this optimized value is consistent with the scant literature on this topic. In their analysis of multi-year measurements of transpiration in loblolly pine, Phillips and Oren (2001) observed that transpiration per unit leaf area was relatively insensitive to increases in leaf area above LAI of approximately 2.5. Iritz and Lindroth (1996) reviewed transpiration data from a range of crop species and found only small increases in transpiration above LAI of 3-4. These authors suggest that the threshold-type responses observed were related to the*

*range of LAI at which self-shading increases most rapidly, therefore limiting increases in transpiration. The resulting model behavior of "flat" transpiration above 2.2 LAI, with gradually decreasing photosynthesis above that value, results in increasing water use efficiency at higher LAI values. Second, the relationship between relative ASW and the modifier of photosynthesis and transpiration predicted a modifier value greater than zero when the relative ASW was zero. This resulted in positive values from photosynthesis and transpiration when the average ASW during the month was zero. In practice, the monthly ASW was rarely zero during simulations, which presents a challenge constraining the shape of the ASW modifier. The priors for the two ASW modifiers (SWconst and SWpower) had ranges that permitted the modifier to be zero. Therefore, additional data are likely needed during very dry conditions to develop a more physically based parameterization. Alternatively, the parameterization of a non-zero soil moisture modifier at zero ASW may be due to trees having access to water at soil depths deeper than the top 1.5 m of soil represented by the bucket in 3-PG. Overall, it is important to view the parameterization presented here as a phenomenological relationship that is consistent with observations from drought and irrigation experiments as well as observations across regional gradients in precipitation.*

3) In response to Reviewer #2, we evaluated how well the model predicts the different experimental types. We now have a figure showing the observed and modeled experimental treatment responses for the data assimilation approaches. In the case of the data assimilation approach that did not include the experimental treatments, the comparison to the observed treatment responses are an independent validation of the model. We found that the data assimilation approach without the experiments predicts the experimental responses reasonably well, except for the $CO_2$ experiment.

*Reflecting this comment, the results section has been modified to the following:*

*In Methods Section 2.4*

*Finally, we compared the predicted responses to experimental manipulation to the observed responses. We focused the comparison on the percentage difference in stem biomass between the control and treatment plots. We used a paired t-test to test for differences between the predicted and observed responses within an experimental type (irrigated, drought, nutrient addition, and elevated CO2). We combined the single and multi-factor treatments for analysis. For the analysis of the nutrient addition studies we only used plots where FR was assumed to be 1 so that we were able to simulate the treatments without requiring the optimization of a site-specific FR parameter.*

*In Methods Section 2.6*

*Fourth, we predicted the treatment plots in the irrigated, drought, nutrient addition (only plots where FR was assumed to be 1), and elevated CO2 plots. As for the Base assimilation, we used a t-test to compare the experimental response between the NoExp assimilation and observed and between the NoExp and Base assimilations. Since the*

*experimental treatments were not used in the optimization, this was an independent evaluation of predictive capacity.*

*In Result section 3.1*

*Furthermore, the response of stem biomass to irrigation (df = 7, p = 0.18), nutrient addition (df = 26, p = 0.29), and elevated CO2 (df = 4, p = 0.43) was not significantly different between the observed and the Base assimilation (Figure 5). The Base assimilation was significantly more sensitive to drought than observed (n = 31, p < 0.001; Figure 5).*

*In Results Section 3.2*

*Excluding the experimental treatments from the data assimilation did not strongly influence the predictive capacity of the model. The RMSE validation plots in NoExp assimilation decreased slightly compared to Base assimilation (21.8 to 18.0 Mg ha-1) while the bias slightly increased (-3.7 to -4.1%)(Figure 4b). Excluding the experimental treatments resulted in a significantly lower response of stem biomass to elevated CO2 than observed (df = 4, p < 0.001; Figure 5). Furthermore, there was a slight negative response of stem biomass to CO2 in the NoExp assimilation because the parameter governing the change in foliage allocation at elevated CO2 (fCpFS700) was unconstrained by observations (Table 6). This led to convergence on the lower bound of the prior distribution (0.5) where foliage allocation decreased with increased atmospheric CO2. The predictions of irrigation, drought, and nutrient addition experiments were not significantly different between the Base and NoExp assimilations (Figure 5).*

4) In response to Anthony Walker's helpful suggestion, we added an additional focus on regional predictions by simulating the regional response to nutrient addition, elevated CO$_2$, and drought. Our new analysis goes beyond the previous analysis by propagating the parameter uncertainty for all HUC12 units in the Southeastern U.S.

*Reflecting this comment, the results section has been modified to the following:*

*In Methods Section 2.6*

*To demonstrate the capacity of the data assimilation system to create regional predictions with uncertainty, we simulated the regional response to a decrease in precipitation, an increase in nutrient availability, and an increase in atmospheric CO2 concentration, each as a single factor change from a 1985-2011 baseline. Each prediction included uncertainty by integrating across the parameter posterior distributions using a Monte-Carlo sample of the parameter chains. Our region corresponded to the native range of loblolly pine and used the HUC12 (USGS 12-digit Hydrological Unit Code) watershed as the scale of simulation. For each HUC12 in the region we used the mean SI, 30-year mean annual temperature, ASW aggregated to the HUC12 level, and monthly meteorology from Abatzoglou (2013) as inputs (Figure 3).*

The SI of each HUC12 was estimated from biophysical variables in the HUC12 using the method described in Sabatia and Burkhart (2014). This SI corresponded to an estimated SI for stands without intensive silvicultural treatments or advanced genetics of planted stock.

To sample parameter uncertainty, we randomly drew 500 samples from the Base assimilation MCMC chain and simulated forest development from a 1985 planting to age 25 in 2011 in each HUC. We choose age 25 as the final age because it is a typical age of harvest in the region. For each sample, we repeated the regional simulation with 1) a 30% reduction in precipitation, 2) FR set to 1, and 3) atmospheric $CO_2$ increased by 200 ppm. Within a parameter sample, we calculated the percentage change in stem biomass at age 25 between control simulation and three simulations with the environmental changes. We focused our regional analysis on the distribution of the percent change in stem biomass.

In Results Section 3.3

Regionally (i.e., the native range of loblolly pines), stem biomass at age 25 ranged from 52 Mg ha-1 to 292 Mg ha-1 with the most productive areas located in the coastal plains and the interior of Mississippi and Alabama (Figure 7a). The least productive locations were the western and northern extents of native range. The width of the 95% quantile interval for each HUC12 unit ranged from 6.2 to 29.8 Mg ha-1 with largest uncertainty located in most the productive HUC12 units and in the far western extent of the region (Figure 7b).

The predicted change in stem biomass at age 25 associated with an additional 200 ppm of atmospheric $CO_2$ over the 1985-2011 levels was similar to the change associated with a removal of nutrient limitation (by setting FR = 1) (Figure 8a,c). The median change associated with elevated $CO_2$ for a given HUC12 unit ranged from 19.2 to 55.7% with a regional median of 21.7% (Figure 8a). The change associated the removal of nutrient limitation ranged from 6.9 to 303.7% for a given HUC12 unit, with regional median of 24.1% (Figure 8b). The response to elevated $CO_2$ was more consistent across space than the response to nutrient addition. The largest potential gains in productivity from nutrient addition were predicted in central Georgia, (Figure 3), the northern extent of the region, and the western extents, areas with the lowest SI (Figure 3).

Stem biomass was considerably less responsive to a 30% decrease in precipitation. The median change in stem biomass when precipitation was reduced from the 1985-2011 levels ranged from -11.6 to – 0.1% for a given HUC12 unit with a regional median of -5.1% (Figure 8c). Central Georgia was the most responsive to precipitation reduction reflecting the relatively low annual precipitation and warm temperatures (Figure 3).

For a given location, the predicted response to elevated $CO_2$ had larger uncertainty than the predicted response to precipitation reduction and nutrient limitation removal (Figure 8c,d,f). The uncertainty, defined as the width of the 95% quantile interval, was consistent across the region for the response to elevated $CO_2$ (Figure 8b). The uncertainty in the

*response to precipitation reduction and nutrient limitation removal was largest in the regions with the largest predicted change (Figure 8df).*

5) Our discussion section is re-worked to reflect the simplified analysis described above.

*The discussion has the following paragraphs*
- *An overall of the findings (same paragraph as reviewed draft)*
- *A paragraph about the hierarchical Bayesian approach (this paragraph is largely from the old methods section, as recommended by Walker)*
- *A paragraph discussing that the CO2 response depended most strongly on the inclusion of ecosystem experiments in the data-assimilation*
- *A paragraph discussing how the model predicted stronger sensitivity to drought than observed and what could be issues with the model*
- *A paragraph discussing why unique parameters were needed for the Duke forest studies to get the CO2 fertilization response correct. (this is a more enriching discussion than the discussion about the 1-stage vs. 2-stage data-assimilation in the previous version*
- *A paragraph about caveats associated with the regional simulations.*

6) In response to comments by both reviewers to justify the set of parameters that were fit, we added six more parameters to the assimilation. We also removed the confusing reference to a sensitivity study of model parameters (the methods describing it were buried in the footnote of a table)

See table 3 for the parameters

7) Sub-sections were added throughout to improve clarity.

*The sub-sections follow the three objectives:*

*1) to present and evaluate a new DA approach that integrates diverse data from multiple locations and experimental treatments with an ecosystem model to estimate the probability distribution of model parameters, 2) to examine how the predictive capacity and optimized parameters differ between an assimilation approach that only uses environmental gradients and an assimilation approach that uses both environmental gradients and ecosystem manipulations, and 3) to demonstrate the capacity of the DA approach to predict, with uncertainty, regional forest dynamics by simulating how forest productivity responds to drought, nutrient fertilization, and elevated atmospheric CO2 across the Southeastern U.S.*

8) We fixed some minor issues with the model structure as follows
   a. The density independent mortality now removes all the biomass of an average individual rather than a proportion of an average individual. This was accomplished by not using the parameter mS (the proportion of an average individual that is lost through turnover) in the density independent mortality calculation.  Since density independent mortality represents random mortality it is more reasonable to not use mS in the calculation.

b.  The model now simulates throughfall experiments directly rather than just reducing rain. Now rain is intercepted by the canopy in the full amount but the rain that enters the soil is reduced when simulating the throughfall experiment. This is a small change that makes the comparison cleaner.

c.  FR is set to 1 in the fertilization studies that added nutrients at regular intervals. Many of these experiments were designed to fertilize to optimal nutrition so the assumption is well grounded and helps reduce the number of site level FR parameters that need to be optimized.

d.  The process error terms are allowed to be a linear function of the prediction.  This allows for the uncertainty to increase with the magnitude of the prediction.  This linear function is applied to stem biomass, GEP, and ET.  It allows for more confidence in predictions of lower values (like winter GEP and ET).

9)  There were improvements to the data assimilation algorithm under the hood that allowed for faster run times and convergence.  The cost function did not change (though we have described the cost function more clearly in the text).

Overall, the updated manuscript is more streamlined (though with more explanation in the methods section) and represents the state-of-the-art for the DAPPER algorithm.

*Specific responses Reviewer #1 (Walker) below*
*Our responses are in italics*

Thomas et al present a data-assimilation (DA) study using constraints from multiple data streams from multiple sites and experiments to optimise parameters in the monthly timestep PG-3 model of loblolly pine production. The study has three specific objec- tives. Stated on lns 170-171, 1) a new regional and hierarchical data assimilation sys- tem with the capacity to assimilate multiple data streams from multiple experiments; stated on ln 179-180, 2) the consequences for parameter estimation and prediction of including or not including ecosystem manipulation experiments (this could be more broadly stated as evaluation of the DA); and stated on ln 181 3) model predictions with the optimised parameter set of forest biomass changes in response to changes in nutrient addition of precipitation. This study is well thought out and implemented, presents a useful advance to the use of DA in ecosystem modelling and forecasting, and will likely be of interest to many readers of Biogeosciences.

My main criticism is that the distinction between the three areas of this study is often not made explicitly throughout the manuscript and consequently the manuscript is not as readable or as clear as it could be.

The majority of my comments are an attempt to help improve the organisation and presentation of the manuscript with the goal that this study will be as widely read and cited as possible.

- With that in mind, I suggest organising the manuscript as much as possible by the three stated objectives. I suggest combining the sentence on lns 179-180 with the sentence on lns 170-171 and explicitly listing the three objectives together. The results and discussion section would benefit from organisation along the lines of the three stated objectives. I suggest breaking each into three subsections, each dealing with one of the objectives. Again the conclusions section should specifically address each objective.

  *We have modified the structure of the manuscript so that the methods, results, and conclusion now have sections that address each of the three objectives.*

Abstract

- It would be good to be specific about who the target audience is for this research. The research straddles a technical field that develops DA but the technique produces a tool at a level of maturity that could be used by foresters. These ultimate end users could be more explicitly targeted.

  *The following text has been added:*

  *"Overall, we 1) demonstrated how three decades of research in southeastern U.S. planted pine forests can be used to develop data assimilation techniques that use multiple locations, multiple data streams, and multiple ecosystem experiment types to optimize parameters and 2) developed a tool for creating future predictions of forest productivity*

*for natural resource managers that are consistent with a rich history of ecosystem research across a region."*

Introduction

- Is a bit long and could a page or so could be cut without loss of content. Paragraphs on lns 82-105 could be combined and reduced in length. The main point is that ecosystem experiments can help to reduce the problem of equifinality in DA.

  *The paragraphs between 82 and 105 were shortened. There is now a single paragraph that is the following:*

  *"Using DA to parameterize ecosystem models with observations from multiple locations that leverage ecosystem manipulation experiments and environmental gradients will allow for predictions to be consistent with the rich history of global change research in forest ecosystems. Ecosystem manipulation experiments provide a controlled environment in which data collected can be used to describe how forests acclimate and operate under altered environmental conditions (Medlyn et al., 2015) and can potentially allow for the optimization of model parameters associated with the altered environmental factor in the experiment. Furthermore, the assimilation of data from ecosystem manipulation experiments may increase parameter identifiability (reducing equifinality (Luo et al., 2009)), where two parameters have compensating controls on the same processes, by isolating the response to a manipulated driver. Observations that span environmental gradients include measures of forests ecosystem stocks and fluxes across a range of climatic conditions, nutrient availabilities, and soil water dynamics. These studies leverage time and space to quantify the sensitivity of forest dynamics to environmental variation. However, covariation of environmental variation can pose challenges separating the responses to individual environmental factors. Overall, assimilating observations from a region that includes environmental gradients and manipulation experiments is a useful extension of prior DA research focused on DA at a single site with multiple types of observations (Keenan et al., 2012; Richardson et al., 2010; Weng and Luo, 2011).*

- The paragraph on lns 108-141 makes some nice points but could be substantially shortened without loss of content. Much of the paragraph is methods like.

  *The paragraph in the comment has been combined with the prior paragraph which is provided above.*

- Weight to rare experiments (mentioned on ln 125) could also apply to rare data types. Later in the paragraph (ln 135-136) the authors state that data of different frequency is a problem in biasing the cost function toward high frequency data, but offer no solution other than a monthly timestep model. Rare data, or low frequency data, could also be given higher weights. Also high frequency data could be summarised at lower frequency.

  *The discussion of the data weighting was removed.*

Methods

Again long and could probably be made more concise. Also the organisation is tough to follow.

- I suggest leading with the observations, the various sites, and measurement campaigns/projects.Many of these are not properly introduced. This will provide a comprehensive introduction to the system and what measurements actually go into this DA system. Observation sites and projects are mentioned on ln 409-410, but these are not introduced and need to be described in the observations section of the methods.

*We moved the section on the observations to the beginning of the methods section.  We structured the paragraph so that it gives an overview of all the measurement campaigns. The observations section (Section 2.1 is as follows)*

*We used thirteen different data streams from 294 plots at 187 unique locations spread across the native range of loblolly pine trees to constrain model parameters (Table 1; Figure 1).  The data streams covered the period between 1981 to 2015. The Forest Modeling Research Cooperative (FMRC) Thinning Study provides the largest number of plots that span the region (Burkhart et al., 1985).  In this study, we only used the control plots that were not thinned.  The Forest Productivity (FPC) Cooperative Region-wide 18 (RW18) study included control and nutrient fertilization addition plots that span the region (134.4 kg ha-1 N + 13.44 kg ha-1 P biannually) (Albaugh et al., 2015).  The PINEMAP study included four locations dispersed across the region that included a replicated factorial experiment with control, nutrient fertilization (224 kg ha-1 N + 27 kg ha-1 P + micronutrients once at project initiation), a throughfall reduction (30% reduction), and fertilization by throughfall treatments (Will et al., 2015).  The SETRES study was located at a single location and included replicated control, irrigation (~650 mm of added water per year), nutrient fertilization (~100 kg N ha-1 + 17 kg P ha-1 with micronutrients applied annually with absolute amount depending on foliar nutrient ratios), and fertilization by irrigation treatments (Albaugh et al., 2004). The Waycross study was a single site with a non-replicated fertilization treatment. The annual application of fertilization focused on satisfying the nutrient demand by the trees was one of the most productive stands in the region (Bryars et al., 2013). These five studies included data streams of stand stem biomass (defined as the sum of stemwood, stembark and branches) and live stem density. Waycross and SETRES included LAI measurements from litterfall traps (Waycross) or estimates from LICOR LAI-2000 (SETRES).  SETRES also included fine root and coarse root measurements.  In the PINEMAP, SETRES, and RW18 studies we only used foliage biomass estimates from the control plots.  We excluded the foliage biomass estimates from the treatment plots because they were derived from allometric models that may not have captured changes in allometry due to the experimental treatment.  We did use LAI measurements from both control and treatment plots where available (SETRES).*

*We also included observations from the Duke FACE study where the atmospheric CO2 was increased by 200 ppm above ambient concentrations. Based on the data presented in McCarthy et al. (2010) the study included six control plots, four CO2 fumigated rings*

*(including the unfertilized half of the prototype), two nitrogen fertilization treatments (115 kg N ha-1 yr-1 applied annually) , and one CO2 by nitrogen addition treatment (fertilized half of prototype). The Duke FACE study included observations of stem biomass (loblolly pine and hardwood), coarse root biomass (loblolly pine and hardwood), fine root biomass (combined loblolly pine and hardwood), stem density (loblolly pine only), leaf turnover (combined loblolly pine and hardwood), fine root production (combined loblolly pine and hardwood), and monthly LAI (loblolly pine and hardwood).*

*Finally, we included two Ameriflux sites with eddy-covariance towers in loblolly pine stands. The US-DK3 site was located in the same forest as the Duke FACE site described above (Novick et al., 2015). The US-NC2 site was located in coastal North Carolina (Noormets et al., 2010). We used monthly gross ecosystem production (GEP; modeled gross primary productivity from net ecosystem exchange measured at an eddy-covariance tower) and evapotranspiration (ET) estimates from the sites. The monthly GET and ET were gap-filled by the site PI. The GEP was a flux partitioned product created by the site PI. The biometric data from the US-DK3 site was assumed to be the same as the first control ring. The biometric data from the US-NC2 site included of stem biomass (loblolly pine and hardwood), coarse root biomass (loblolly pine and hardwood), fine root biomass (combined loblolly pine and hardwood), stem density (loblolly pine only), leaf turnover (combined loblolly pine and hardwood), and fine root production (combined loblolly pine and hardwood).*

- I found section 2.3 very difficult to follow. I'm not expert on DA mathematical meth- ods but I have a reasonable conceptual handle on DA, and yet I was lost in the first paragraph. I also ran this section by a colleague who is expert in the mathematics underpinning DA and they agreed that this sections needs to be clearer. Their key criticism was that they could not see the derivation of Eq 7, perhaps the authors could add the derivation to an appendix. And that it is not clear how the MCMC was used to sample Eq 7. A clear description of the details of the MCMC procedure is necessary, along with the presentation of the cost function. Also the first term on the righthand side of Eq 7 is not the same as the righthand side of Eq 1, is this deliberate? And E is never defined.

  I strongly suggest reworking section 2.3 of the methods to be extremely clear about the DA process and how it was implemented. Start with a clear description of the goals of the DA – state estimation and estimation of parameter distributions. Then describe all the various sources of uncertainty and how the method accounts for them. Then take the reader step by step through the method. Perhaps a diagram would be useful. The following comments are an attempt to provide examples of where confusion arises but they are in no way comprehensive. The sentence on lns 281-283 is more or less stating the the same thing as the sentence on lns 284-285. I suggest fusing these together. Is the reference to a "latent model" really necessary, it is confusing with the mathematical model. Would "true" system states and fluxes convey the same meaning? Do not try to justify the method in comparison with previous methods (e.g. lns 286-291), in the methods this just confuses the description and this can be argued in the discussion. On lns 291-293, this is state estimation right? That's fine but is it really the focus of your method? None of the three stated objectives are for state estimation. How exactly was estimation of the latent state or flux the first step in the process when it includes the optimised parameters etc as described on lns 296-298? Seems like the statement on ln 306-308 should come before the previous paragraph.

*We cleaned up the description of the cost function per the reviewer recommendation. (see Supplement to the review)*

*We used a hierarchal Bayesian framework to estimate the posterior distributions of parameters, latent states of stocks and fluxes, and process uncertainty parameters. The latent states represented a value of the stock or flux before uncertainty was added through measurement. The approach was as follows.*

*Consider a stock or flux (m) for a single plot (p) at time t ($q_{p,m,t}$). $q_{p,m,t}$ is influenced by the processes represented in the 3-PG model and a normally distributed model process error term,*

$$q_{p,m,t} \sim N(f(\boldsymbol{\theta}, FR_p), \sigma_m) \qquad\qquad \textit{Equation 1}$$

*where $\boldsymbol{\theta}$ is a vector of parameters that are optimized, $FR_p$ is the site fertility, and $\sigma_m$ is the model process error. Not shown are the vector of parameters that were not optimized (Supplemental Material Table 1), the plot ASW, an array climate inputs, and the initial conditions because these are assumed known and not estimated in the hierarchical model. The process error assumed that the error linearly scales with the magnitude of the prediction:*

$$\sigma_m^2 = \gamma_m + \rho_m f(\boldsymbol{\theta}, FR_p) \qquad\qquad \textit{Equation 2}$$

*While the structure of the Bayesian model allowed for all data streams to have process uncertainty that scales with the prediction, in this application we only allowed stem biomass, GEP, and ET process uncertainty to scale because they had large variation across space (stem biomass) and through time (i.e., there should be lower process uncertainty in the winter when GEP is lower). For the other data streams, the linear scaling term was removed by fixing $\rho_m$ at 0.*

*$FR_p$ did not have an explicit probability distribution. Rather the probability density evaluated to 1 if the plot was not fertilized, thus causing $FR_p$ to be estimated from SI and MAT (Supplemental Material Equation 15), or if it was a fertilized plot and has an $FR_p$ equal or higher than that of its non-fertilized control plot. The probability density evaluated to 0 if the estimated $FR_p$ in a fertilized plot was less than the $FR_p$ in the control plot or $FR_p$ awas not contained in the interval between 0 and 1.*

$$FR_p \sim \begin{cases} 1 \text{ if non-fertilized, } FR_p \geq 0, \text{ and } FR_p \leq 1 \\ 1 \text{ if } FR_p = 1 \text{ and fertilization levels are assumed to remove nutrient deficiencies} \\ 0 \text{ if } FR_p < 1 \text{ and fertilization levels are assumed to remove nutrient deficiencies} \\ 1 \text{ if fertilized but levels are not assumed to remove deficiencies and } FR_p \geq FR \text{ of control plot} \\ 0 \text{ if fertilized but levels are not assumed to remove deficiencies and } FR_p < FR \text{ of control plot} \\ 0 \text{ if } FR_p < 0 \text{ or } FR_p > 1 \end{cases}$$

*Equation 3*

*Our model included the effect of observational errors for measurements of stocks and fluxes. For a single stocks or flux for a plot at time t there is an observation ($y_{p,m,t}$). The normally distributed observation error model was:*

$$y_{p,m,t} \sim N(q_{p,m,t}, \tau^2_{p,m,t}) \qquad \text{Equation 4}$$

*where $\tau^2_{p,m,t}$ represented the measurement error of the observed state or flux. By including the observational error model, $q_{p,m,t}$ represented the latent, or unobserved, stock or flux. The variance was unique to each observation because it was represented as a proportion of the observed value. The $\tau^2_{p,m,t}$ was assumed known (see Table 2) and not estimated in the hierarchical model (Table 2).*

*The hierarchical model required prior distributions for all optimized parameters, including the parameters for the 3-PG model ($\theta$), $FR_p$, and the process error parameters. The prior distributions for $\theta$ are specified in Table 3. Some parameters were informed by previous research in loblolly pine ecosystems while other parameters were 'non-informative' with flat distributions (termed 'vague' in Table 3). The prior distributions*

*for the process error parameters were non-informative and had a uniform distribution with upper and lower bounds that spanned the range of reasonable error terms.*

$\gamma_m \sim U(0.001, 100)$    *Equation 5*

$\rho_m \sim U(0,10)$   *Equation 6*

*By combining the data, process, and prior models, our joint posterior that includes all thirteen data streams, plots, months with observations, and fitted parameters was*

$$p(\boldsymbol{\theta}, \boldsymbol{\gamma}, \boldsymbol{\rho}, \boldsymbol{q} | \boldsymbol{y}, \boldsymbol{\tau}, priors) \propto$$

$$\prod_{p=1}^{P} \prod_{m=1}^{M} \prod_{t=1}^{T} N(q_{p,m,t} | f(\boldsymbol{\theta}, FR_p), \gamma_m + \rho_m f(\boldsymbol{\theta}, FR_p))$$

$$\prod_{p=1}^{P} \prod_{m=1}^{M} \prod_{t=1}^{T} N(y_{p,m,t} | q_{p,m,t}, \tau_{p,m,t}^2)$$

$$\prod_{p=1}^{P} p(FR_p) \prod_{f=1}^{F} p(\theta_f) \prod_{m=1}^{M} p(\gamma_m) \prod_{m=1}^{M} p(\rho_m)$$

*Equation 7*

*where bolded components represent vectors, P is the total number of plots, M is the total number of data streams, T is the total months with observations, and F is the total number of 3-PG parameters that are optimized.*

*We numerically estimated the joint posterior distribution using the Monte-Carlo Markov Chain – Metropolis Hasting (MCMC-MH) algorithm (Zobitz et al., 2011). This approach has been widely used to approximate parameter distributions in ecosystem DA research (Fox et al., 2009; Trudinger et al., 2007; Williams et al., 2005; Zobitz et al., 2011). Briefly, the algorithm proposes new values for the model parameters, uncertainty parameters, latent states, and FR. The proposed values were generated using a random draw from a normal distribution with a mean equal to the previously accepted value for that parameter and standard deviation equal to the parameter-specific jumping size. The ratio of proposed calculation of Equation 7 to the previously accepted calculation of Equation 7 was used to determine if the proposed parameters are accepted.  If the ratio was greater than or equal to 1 the proposed values were always accepted.  If the ratio was less than 1, a random number between 0 and 1 was drawn and the proposed values*

*are accepted if the ratio was greater than the random number. This allowed less probable parameter sets to be accepted, thus sampling the posterior distribution. We adapted the size of the jump size for each parameter to ensure the acceptance rate of the parameter set was between 22% and 43% (Ziehn et al., 2012) by adjusting the jump size if the acceptance rate for a parameter is outside the 22 – 43% range. All MCMC-MH chains were run for 30 million iterations with the first 15 million iterations discarded as the burn-in. Four chains were run and tested for convergence using the Gelman–Rubin convergence criterion, where a value for the criterion less than 1.1 indicated an acceptable level of convergence. We sampled every 1000th parameter in the final 15 million iterations of the MCMC-MH chain and used this thinned chain in the analysis described below. The 3-PG model and MCMC-MH algorithm were programed in FORTRAN 90 and used OpenMP to parallelize the simulation of each plot within an iteration of the MCMC-MH algorithm.*

- Section 2.4 jumps around between objectives. Some text would fit better in section 2.3, for example lns 408-428. Text on lns 454-461 would be better organised if it were to follow the text on 430-444, then the regional simulations can be presented afterwards.

  *We reorganized as suggested by the reviewer. Section 2.1 is the observations, Section 2.2 is the Ecosystem Mode, Section 2.3 is the data assimilation method, Section 2.4 is the data assimilation evaluation, Section 2.5 is the Sensitivity to the inclusion of ecosystem experiments, Section 2.6 is the Regional predictions with uncertainty.*

- I suggest defining sections 2.3, 2.4, and an additional 2.5 to be organised by the three stated objectives.

  *We reorganized as suggested by the reviewer. Section 2.1 is the observations, Section 2.2 is the Ecosystem Mode, Section 2.3 is the data assimilation method, Section 2.4 is the data assimilation evaluation, Section 2.5 is the Sensitivity to the inclusion of ecosystem experiments, Section 2.6 is the Regional predictions with uncertainty.*

- Also, while commonly used by the modelling community, I do not agree that you can run "experiments" with models. Models make predictions from a specific set of mathematical hypotheses and defined scenarios. An experiment is designed to test predictions and discriminate among hypotheses.

  *We removed the 'experiments' language*

Results

- Why were only 31 parameters optimised, can you describe why this set were chosen from the total 46?

*In the revised manuscript, we included more parameters that were optimized (six more). The eight parameters that were not optimized did not have specific data to use as a constraint (leaf boundary layer, conductance, canopy light extinction coefficient, etc).*

- Technically the parameters are not "sensitive" (ln 480), it is the model output that is sensitive to the parameter. "Influential" would be a better adjective to describe the parameters.

    *To simplify the analysis and reduce the density of the manuscript we removed the sensitivity study and the reference to it in the text.*

- Lns 486 & 488 variability is described as being reduced but no data are provided. Can you quantify these statements. There are many statements like this throughout the results and they ought to be quantified (e.g. lns 502, 508). Also on 508, is mean correct, isn't this the median of the parameter distribution?

    *We added a column to the table that is the ratio of the size of the posterior 99% credible interval to the size of the prior 99% confidence interval. This ratio illustrates how the uncertainty is reduced by the data assimilation.*

- Some kind of visual representation of the data in table 5 would be useful.

    *Supplemental Material Figure 1 shows the PDF of the prior and posterior*

- Ln 492 what do you mean strong priors? Well defined from measurements and litera- ture with low variance? Could you quantify this?

    *We removed this language from the manuscript to reduce confusion*

- Lns 494 the process uncertainty parameters are mentioned here and in the methods, but results are barely presented (only in the supplement) and are not discussed, or not that I noticed. This is a very interesting concept and I would like to see these data pre- sented a little more and at least a little discussed. What kind of impact does including these parameters have on the optimised parameter distributions? I understand you are already presenting a lot, but this is fairly novel as far as I'm aware and is of interest.

    *We added a small discussion of the process error parameters to the discussion section*

- Figure 10 and 11 would be more in keeping with your stated goal of forecasting on lns 65-68 if you removed the b panels in both plots. If you think that the parameter estimates when including the data from the manipulations gives a better estimate of those parameters then the data in panels b are not particularly useful for forecasts. In my view, and as stated on line 67 & 68 "provide information on both the expected future state of the forest and the probability distribution of those future states", the final figures would be much stronger if the probability distribution of the future states shown on the a panels were represented on the b panels.

*We combined the Figure 10 and 11 into a single figure that has the median prediction on the left side and the uncertainty on the right side. This allows the figures to represent the forecasting capacities of the data assimilation approach. The paragraph is as follows:*

*Our hierarchal approach (Equation 7) was designed to partition uncertainty that is attributable to uncertainty in parameters, model process, and measurements (Hobbs and Hooten, 2015). Previous forest ecosystem DA efforts have either focused on parameter uncertainty, by using measurement uncertainty as the variance term in a Gaussian cost function (Bloom and Williams, 2015; Keenan et al., 2012; Richardson et al., 2010) or on total uncertainty by directly estimating the Gaussian variance term (Ricciuto et al., 2008). The latter combines measurement uncertainty and process uncertainty into the same parameter and is unable to be used for developing prediction intervals, as prediction intervals only include parameter and process errors (Dietze et al., 2013; Hobbs and Hooten, 2015). Our approach allows the estimation of the probability distribution of forest biomass before uncertainty is added through measurement. Considering that the method of assimilation can potentially have a large influences on posterior parameter distributions (Trudinger et al., 2007), future research should focus on comparing the hierarchal approach presented here to other approaches by using the same data constraints with alternative cost functions.*

- While it is interesting to show the consequences for prediction of inclusion of manipulations or not, and the opposite sign of the change in predictions when water and nutrient manipulations are included, you already show this in Figures 6 & 7. If you want to keep the b panels in 10 and 11 I suggest you add them as extra panels to figures 6 & 7, showing the absolute delta (or similar) from the simulations that include the manipulation delta. This will allow you to address the question: what are the consequences of not including data from manipulations? Without confounding the predictions from the most appropriate DA product for the scenarios tested. Also, the scale ought to be the same for the data presented in Figs 10 and 11.

    *We cut panel b from these figure.*

- Was CO2 change included in the above projections of removal of nutrient limitation and precipitation reduction? Furthermore, it seems you have included data from water manipulation experiments, nutrient manipulation experiments, and CO2 manipulation experiments. But you have only made projections for nutrient and precipitation change. Why not CO2 change? CO2 projections would complete the study.

    *We added a +200 ppm simulation to the set of regional predictions. The predicted regional changes are for +200 ppm, -30% precipitation, and removal of nutrient limitation. The uncertainty for each prediction is shown. This changed the description of the regional results to be the following:*

*Regionally (i.e., the native range of loblolly pines), stem biomass at age 25 ranged from 52 Mg ha-1 to 292 Mg ha-1 with the most productive areas located in the coastal plains and the interior of Mississippi and Alabama (Figure 7a). The least productive locations were the western and northern extents of native range. The width of the 95% quantile interval for each HUC12 unit ranged from 6.2 to 29.8 Mg ha-1 with largest uncertainty located in most the productive HUC12 units and in the far western extent of the region (Figure 7b).*

*The predicted change in stem biomass at age 25 associated with an additional 200 ppm of atmospheric $CO_2$ over the 1985-2011 levels was similar to the change associated with a removal of nutrient limitation (by setting FR = 1) (Figure 8a,c). The median change associated with elevated $CO_2$ for a given HUC12 unit ranged from 19.2 to 55.7% with a regional median of 21.7% (Figure 8a). The change associated the removal of nutrient limitation ranged from 6.9 to 303.7% for a given HUC12 unit, with regional median of 24.1% (Figure 8b). The response to elevated $CO_2$ was more consistent across space than the response to nutrient addition. The largest potential gains in productivity from nutrient addition were predicted in central Georgia, (Figure 3), the northern extent of the region, and the western extents, areas with the lowest SI (Figure 3).*

*Stem biomass was considerably less responsive to a 30% decrease in precipitation. The median change in stem biomass when precipitation was reduced from the 1985-2011 levels ranged from -11.6 to – 0.1% for a given HUC12 unit with a regional median of - 5.1% (Figure 8c). Central Georgia was the most responsive to precipitation reduction reflecting the relatively low annual precipitation and warm temperatures (Figure 3).*

*For a given location, the predicted response to elevated $CO_2$ had larger uncertainty than the predicted response to precipitation reduction and nutrient limitation removal (Figure 8c,d,f). The uncertainty, defined as the width of the 95% quantile interval, was consistent across the region for the response to elevated $CO_2$ (Figure 8b). The uncertainty in the response to precipitation reduction and nutrient limitation removal was largest in the regions with the largest predicted change (Figure 8df).*

Additional points

- I think the title would benefit from the addition of "Loblolly Pine".

  *Added to title*

- Ln 50 Duke FACE experiment had 4 replicate plots, so where does the 5 come from on this line. An additional plot from the unreplicated prototype?

*We removed the language from the abstract and later in the text we clarified that the replicated prototype was used (per the data reported in McCarthy et al. 2010)*

- Ln 48 – 50 the sentence on this line would help flow if it were before the preceding sentence.

  *Revised*

- Ln 65 I don't think I would classify the three areas mentioned in the previous sentence as tools. They are more than tools, they are also knowledge.

  *Removed the word 'tools' so that the sentence references the previous sentence terminology ('sources of information')*

- Ln 67 What do you mean by "based on" here. Can probably delete. Also while I think your methods could be used for "forecasting" you don't really use the method in that sense.

  *Removed the clause that contained 'based on'*

- Ln 73 insert "can" in between "that generate"

  *Fixed in text*

- Ln 85 86 "carbon allocation and turnover" This is worded a little awkwardly

  *Removed awkward language from text*

- Ln 97-99 awkward way to start a paragraph.

  *Paragraph was removed during the shortening of the introduction*

- Ln 111 suggest replacing "important" with "useful" or something more descriptive

  *Changed to 'useful'*

- Ln 155-157 suggest replacing "nutrients" with "nutrient addition". Also suggest removing hyphens.

  *Changed in text*

- Ln 162-163 Awkward

  *Removed 'available' to make less awkward*

- Ln 171 Again I think you need to call out loblolly pine here

*Changed in text when revision the statement of objectives*

- Ln 175 The authors chosen acronym, in my view, somewhat undersells what they are doing. The DA method is hierarchical and considers data from multiple sites and of multiple different types. The acronym gives not indication of this and suggests that the DA method is only suitable for Pine Plantations. Of course it is the authors' choice though.

  *Thank you for the suggestion to broaden the acronym. We kept the same acronym but changed the words to "Data Assimilation to Predict Productivity for Ecosystems and Regions" to emphasize the multi-site aspect of the DA.*

- Ln 307 insert "considered" between "was a"

  *Sentence was modified during revisions*

- Ln 446 replace "regional" with "region"

  *Changed in text*

- Ln 522-524 I'm not sure what you mean here, could you clarify?

  *Sentence removed during the revisions*

- Ln 528 delete "a"

  *Done*

- Ln 576 replace "detangling" with "disentangling"

  *Done*

- Ln 582 I think "synthesised" would be a better word to use than "organised"

  *Done*

- Ln 591-591 I take your point about equifinality but can you really say this if predictions were not improved in some way? Just a thought. Is there a way that you can be sure that the mechanisms were correctly distinguished?

  *We removed this sentence during revisions*

- Ln 633-634 Agreed, but did your method strictly weight the data? Wasn't it more that the hierarchical method gave priority to the CO2 manipulation data?

  *We removed this sentence during revisions*

- Ln 646 replace "than" with "that"

  *Done*

- Ln 656 quantify this statement

  *We removed this sentence during revisions*

- Ln 662-663 this was news to me when I read this sentence. I think this would become clearer once the methods can be clarified as suggested above.

  *We clarified in the method section. The method section more completely describes assumptions of the site index estimation. The following text was added to Section 2.6 in the methods:*

  *The SI of each HUC12 was estimated from biophysical variables in the HUC12 using the method described in Sabatia and Burkhart (2014). This SI corresponded to an estimated SI for stands without intensive silvicultural treatments or advanced genetics of planted stock.*

- Ln 668 suggest changing "prior" to "previous", just to maintain the meaning of prior in the Bayesian sense.

  *Done*

- Ln 673 you do not show any data on covariation of parameters.

  *We removed this language*

- Ln 676-680 I like this statement, makes a lot of sense. But is it most appropriate here? This point should be made clearly in the methods.

  *Moved to methods*

- Ln 685 suggest deleting "Multivariate Constructed Analogs (MACA)" it is not needed.

  *Deleted*

- Ln 692-697 This is a good point but I'm curious why the change in biomass in response to precipitation reduction was small given the large change in parameter values when water manipulations were included in the DA. Can you try to explain this based on the process hypotheses embedded in the model.

  *We cut this sentence during revisions*

- Ln 698 replace "reduced" with "reduction" Ln 707 insert "as a function of"

*Done*

- Ln 719 insert space in "fromadditional"

*Done*

- Ln 760 While I'm sure the methods and tools developed by this study could be used for ecological forecasting, strictly speaking this study is not ecological forecasting. The third objective, which concerns optimised model predictions, is a scenario analysis rather than a forecast.

*We removed the term 'ecological forecast' from the sentence and changed to:*

*DA is increasingly used for developing predictions from ecosystem models that include uncertainty estimation, due to its ability represent prior knowledge, integrate observations into the parameterization, and estimate multiple components of uncertainty, including observation, parameter, and process representation uncertainty (Dietze et al., 2013; Luo et al., 2011b; Niu et al., 2014).*

- Ln 769 no need to cite Medlyn et al 2015 here

*Removed citation*

*Specific responses Reviewer #2*
*Our responses are in italics*

Quinn Thomas et al. present a model-data fusion, or data assimilation, study that gathers 35 years of carbon cycle-related observations and manipulation experiments taken in Loblolly Pine ecosystems in the Southeastern US to optimize parameters of the 3-PG model within their new framework DAPPER. The authors examine the ability of the observations to constrain model parameters using a number of approaches for assimilating the different types of data, and they further examine the differences in model behavior/sensitivity and change in biomass stocks across the southeastern US as a result of the different experiments.

The authors have carried out an impressive and exhaustive collection of data for con- straining the 3-PG model in this study. This, and their investigation into different approaches for assimilating different types of data, in particular manipulation experimental data, make this study a noteworthy contribution to model–data assimilation literature in forested ecosystems, and therefore I would recommend publication in Biogeosciences. However, as it stands the manuscript is quite long and dense, which is understandable given the amount of detail that is required to present such a wide array of data and experiments. This being said, I recommend that the authors try to edit the article following some of the suggestions below (and their own views) to improve the clarity and readability of the text before this article is published.

- Overall, the objectives and key points of this study can get lost in the text. I think a few more sub-sections in the main text and supplementary, references and links between sections would help the reader to better follow and absorb the necessary amount of detail presented in the manuscript. I would also find it useful if the authors posed a few key scientific questions to help them highlight the main messages of the study.

  *We clarified the last paragraph of the introduction to directly state the three objectives of the study. We also added section to the Methods, Results, and Discussion that parallel the objectives*

  *The objectives paragraph is as follows:*

  *Using loblolly pine plantations across the southeastern U.S as a focal application, our objectives are to 1) present and evaluate a new DA approach that integrates diverse data from multiple locations and experimental treatments with an ecosystem model to estimate the probability distribution of model parameters, 2) examine how the predictive capacity and optimized parameters differ between an assimilation approach that only uses environmental gradients and an assimilation approach that uses both environmental gradients and ecosystem manipulations, and 3) demonstrate the capacity of the DA approach to predict, with uncertainty, regional forest dynamics by simulating how forest productivity responds to drought, nutrient fertilization, and elevated atmospheric CO2 across the Southeastern U.S.*

- Some sections in the methods could do with more explanation for why certain approaches were used (see comments below) or better links to the supplementary material, as I have just mentioned.

  *See comments below for response*

- The introduction and discussion are quite long and this can prevent some of the key points from being highlighted. I suggest the authors try to cut down the text where they see fit, including some sentences that essentially are repetitions of earlier statements.

  *We cut the introduction and removed paragraphs*

- The paragraphs in the results section could be separate sections with sub-headings in order to guide the reader, while at the same time the results could benefit from stronger links between each section, especially before line 522, in particular comparing the between the 1st and 2nd stages, or the different 2-stage approaches with the 1- stage approach. At the moment, the results section before line 522 is a bit fragmented, making it harder to weave together a coherent story that brings out the key points.

  *We added sub-sections to the results section*

- Reading this manuscript I found myself asking: What do you expect from each experiment/approach? What will you gain/lose? Which approach is the right approach, going forward? These questions were largely answered in the discussion, and therefore I have made a suggestion below that perhaps some of the results and discussion could be merged within the sub-sections suggested above. This is a personal style issue however.

  *We hope that updated analysis and discussion section helps answer these questions more clearly. There are now sub-sections in the results and discussion that help provide continuity between the sections.*

- Finally, the authors may consider cutting other sections of the discussion that are not fully pertinent to the results as the paper is already quite full of detail. I would like to stress that despite this suggestion I did find the discussion to be interesting and comprehensive, but I would like to see the key messages highlighted more and am concerned the length of the paper may overwhelm the reader.

  *We have cut out the paragraphs that aren't directly related to the results. These include the paragraph about the connections to the Community Land Model and the paragraph about the connections to sap-flux measurements*

Introduction
- Line 97: "relative contribution of each environmental control should be separated in order to correctly parameterize the sensitivity to changes in the environment". I agree to some extent but this is very hard to do and should we be separating each environmen- tal control, as the interaction between different environmental changes may produce different outcomes than if each were treated separately? I would be interested to hear the authors thoughts on this and what they think the impact of assimilating manipulation experiments data separately has on their results.

*Per reviewer #1 comment to shorten this paragraph, this sentence is now removed from the manuscript.*

- Line 124-128: See previous studies Wutzler and Carvalhais (2014) and Section 2 of MacBean et al. (2016) for further discussion on debate of how to deal with the issue of weighting to account for the number of observations and/or using a multi-stage assimilation approach to address challenges of assimilating a diverse set of observations. Both issues are the subject of debate in the literature. On the issue of weighting by the number of observations, from a mathematical standpoint there would be no need if the error covariance matrix is properly characterized; however, this is difficult to achieve in practice. Similarly, a joint or simultaneous assimilation, in which all observations are assimilated together, is mathematically more rigorous as the error covariance between the observations can be properly taken into account. I appreciate that you have dis- cussed the benefit of weighting by the type of data in the discussion, but this debate in the literature (for and against weighting, due to the abovementioned reasons) should perhaps be referred to more clearly in this study.

*Per reviewer #1 comment to shorten this paragraph and review #2 comment that the discussion lacks of the data weighting lacks precision, we cut this discussion.*

- Line 129: It is true of course that to constrain changes in biomass monthly time-scale models are sufficient, but note that monthly time-scale models are not the only way to overcome computational challenges associated with inverting a complex ecosystem model. There are sophisticated yet simple algorithms that dramatically improve the sampling of parameter space in a limited number of iterations. See the work of Jasper Vrugt: https://scholar.google.co.uk/citations?user=zkNXecUAAAAJ&hl=en&oi=ao

*We cut the discussion about monthly time-step models while shortening the paragraph but will definitely look more closely into the work by Vrught. Thanks for highlighting!*

Methods

- Section 2.1 It would be good if you could refer to references and/or relevant sections in the Supplement in Section2.1 to depict between standard characteristics of the 3PG model specific additions or alternative choices you made and (and to explain why you made those choices). For example:

*Added subsections to the Supplemental Material and added the references to Supplemental Material to the main text*

- Line 201-202: Was this additional function based on a published study?

*The function was developed as part of this study*

- Line 209: Is the site-index a new addition to the model that you developed? If so, from where?

  *The text now reads:*

  *For unfertilized plots, we used site index (SI), a measure of the height of a stand at a specified age (25 years), to estimate FR. This approach is in keeping with previous efforts (Gonzalez-Benecke et al., 2016; Subedi et al., 2015)*

- Lines 218-220: Why did you remove the dependence of total root allocation on FR for the DA study?

  *We removed the dependence of total root allocation on FR because we separated root allocation into the coarse and fine roots. Therefore, the previous function was not applicable. Future studies should investigate how best to build this function back in and ask whether we currently have the observational constraints to parameterize it.*

- Line 229-231: A reference for or further explanation of this modification would be good here. –

  *Added text*

- Line 245: "implicit irrigation in very dry conditions." Is this a realistic feature of these sites? How does this affect the results? Especially for the water availability manipulation experiments.

  *We added text explain how this assumption could influence the results. "This assumption may cause the model to be less sensitive to low soil availability but the optimized parameterization may compensate. "*

- Line 250: do you mean to say "mean monthly GPP"?

  *GPP was a sum for each month so 'monthly GPP' is correct. Mean monthly GPP might imply that multiple months are averaged.*

- Line 251-252: How did you select the 31 parameters to be optimized?

  *In the revised manuscript, we included more parameters that were optimized (six more). The eight parameters that were not optimized did not have specific data to use as a constraint (leaf boundary layer, conductance, canopy light extinction coefficient, etc).*

  *The paragraph in the results section now reads as follows:*

*Our multi-site, multi-experiment, multi-data stream DA approach (Base assimilation) increased confidence in the model parameters (Table 5). Averaged across parameters, the posterior 99% quantile range from the Base assimilation was 60% less than the prior range. The largest reduction in parameter uncertainty was for the parameters associated with light-use efficiency (alpha) and the conversion of GPP to NPP (y), which on average had ranges that were 85% lower in the posterior than the prior. Parameters associated with allocation and allometry had a 63% reduction in the range while parameters associated with mortality processes had 70% reduction in the range. Parameters associated with environmental modifiers had the least reduction in the range with a 40% decrease. In addition to the parameters associated with the 3-PG model, the model process error parameters for each data stream were well constrained with large reductions in the range (> 99% decrease; Supplemental Material Table 2)*

- Table 1: Please can you give the equation for how the sensitivity is calculated? Also, please could you explain why there is both a number and "vague" given for the uncertainty of some parameters? If "vague", please can you detail how you defined the prior uncertainty/ranges in the text?

*We cut out the sensitivity analysis and added more parameters to the optimization.*

- Finally, I appreciate you have a lot of information to con- vey and the tables are large, but it might be good to have all optimized parameters here and just indicated which ones are referred to in the discussion.

*We expanded the table to include all optimized parameters*

- As a general comment, it is hard to find some of the information you refer to in the Supplement (e.g. the other optimized parameters you refer to in the caption of Table 1). Please could you split the Supplement into numbered/indexed sections and then refer specifically to the relevant section to help the reader?

*We added section divisions to the supplemental material*

- Line 255-265: How did you initiate the biomass pools? Based on site-level data for the start of the simulation period? Please detail with references. If no site data were available, how sensitive were your DA experiments on the method used to initiate the biomass pools? Later note: I see you have addressed this in Section 2.4. It might be useful to refer to that section here so the reader is not questioning this in this section.

*We moved the text on the initialization described to the section on the model description*

- Section 2.2 Table 2: Last column – Table 3 instead of Table 4. Also, please could uou explain, or give references, for why the SD for observations sometimes varied between 10% and 2.5% of the observation.

*To reduce confusion, we used 10% for LAI observations. Future applications of the method can focus more on the influence of data uncertainty on parameter estimates.*

- Section 2.3 Equation 4: Please explain why you picked a uniform distribution between 0.001 and 100?

  *We added text to state that the bounds of 0.001 to 100 were designed allow the priors to be vague. The bounds include reasonable ranges of standard deviation parameters.*

- Lines 348-349: Please explain why (only) 3 MCMC chains were run? Was a convergence metric such as R-hat used?

  *We re-ran our optimization with the updates described at the top of the response. We ran 4 chains and used the Gelman R criteria to test for convergence. The methods section now includes the following text:*

  *Four chains were run and tested for convergence using the Gelman–Rubin convergence criterion, where a value for the criterion less than 1.1 indicated an acceptable level of convergence.*

- Section 2.4 Lines 398-399: Although I understand the reasoning that these sites are close together and the most data rich, I don't understand why you lump the Duke $CO_2$ enrichment site with DK3 and NC2 in the 1st stage when you stated that you wanted to test the influence of the $CO_2$ fertilization – why not just test the Duke $CO_2$ enrichment site by itself in the 1st stage and the remaining sites/plots in the 2nd stage to answer this question?

  *Addressing this comment was the one of the primary reasons that we re-ran and simplified our analysis. Our updated analysis removed the need for a 2-step analysis. (see beginning of this response for more info).*

- Further to the above point, I appreciate the extra experiments to understand the influence of the $CO_2$ fertilization on the posterior parameters, and the further experiments to determine the influence of the water treatments and nutrient addition. But how dependent are your results on which type of observation and/or treatment is assimilated in the 1st stage vs 2nd stage? Would the results different if you reverse the stages you have in your current set-up? Again, see Wutzler and Carvalhais (2014) and/or MacBean et al. (2016) who discuss these issues (as well as the issue of the weight of different types of data, as you discuss below. A pseudo-test with synthetic observations would have been useful prior to assimilating real data to determine whether the exact set-up of a 2-stage assimilation is sensitive to the order of observation assimilation as well as to confirm if the assimilation system is able to constrain the parameters to their correct values.

  *Our updated analysis removed the need for a 2-step analysis. (see beginning of this response for more info)*

- Lines 430-465: While the tests and approaches put forward here are interesting, the text is dense. Any efforts the authors could make to simplify the description of the experiments and simulations performed (perhaps with the use of a table and simula- tion/experiment code names?) would likely help the reader.

  *We reorganized and clarified this text in response to this comment and comments from Reviewer 1. We have a Base (all plots, three unique parameters for the Duke site), NoExp (no experimental treatments, three unique parameters for the Duke site) and NoDkPars (all plots, no unique parameters for the Duke site)*

- Lines 467-475: The cross-validation exercise presented here is a useful one. Was a similar test used to assess the validity of the posterior distributions of the manipulation experiments, even though there are fewer sites?

  *We added a cross validation of the experiments treatments. We now include optimized parameter set that did not include the experimentally treated plots in the assimilation. This parameter set is now used to predict the experimental treatments.*

Results
- Line 480-484: Description of the sensitivity analysis and choice of parameters should be in the methods. Was this a one-at-a-time sensitivity analysis or a full global method? What is the justification for using this approach versus an existing global sensitivity analysis that accounts for correlations between parameters and explores the whole parameter space (unless I have misunderstood what was done)?

  *We cut the reference to the sensitivity analysis*

- Why did you fix the light extinction coefficient as opposed to the quantum yield parameter?

  *We fixed the light extinction coefficient because it was more known than the canopy quantum yield.*

- Supplemental Table 3 and Table 5: As mentioned above I would suggest having all the optimized parameters in one table. I would also suggest putting the prior min/max in Table 5 even though it might mean having an extra line/column per parameter and taking this information out of table 1 so it is easier to see how well the optimization has constrained the parameters.

  *We moved all parameters to the table in the main text and added the range uncertainty in the priors to the same table*

- Finally, I would suggest splitting up the parameter tables into the sections you refer to in the text, e.g. "temperature sensitivity of quantum yield" or "physiological parameters" etc. This will make it easier for the reader to refer to the tables when reading the text.

*Done*

- Which experiment do the supplemental figures correspond to? The "ALL" experiment? This should be detailed.

  *The assimilation approaches have been renamed and clarified in the supplemental figures.*

- Are you talking about the 1st stage experiment in the first paragraph of the results? If so, it would be good to specify this, and I would further suggest splitting the results into sections to more easily guide the reader.

  *We clarified by using the names of the data assimilation approaches. Our results section is better organized in response to review 2.*

- Do you discuss DK+NC2-fert in the results, or have I missed it? Perhaps more needed on the 1-stage versus 1st and 2nd stages before you discuss the experiments with and without nutrient and water addition (i.e. before line 522)?

  *Our updated analysis did not require the 2-stage approach so we no longer need to report the DK+NC2-fert results. This helps simplify the description of the results.*

- Figure 5 comes before Figure 4 in the text – switch around?

  *Fixed in text*

- Lines 507-515: I am a bit confused by the sentence "The two-stage assimilation was critical for constraining the CO2 quantum yield enhancement parameter (Calpha700)" as you then go on to say (and show, in Figure 5) that the 1 stage resulted in a narrower uncertainty interval? I guess you mean that despite the higher 95% confidence interval, the 2-stage approach results in a more realistic parameter value but I am not at all sure on that? Please could you clarify this in the text?

  *Paragraph was modified in the revisions*

- Line 517: I would suggest putting the names of the soil fertility parameters in brackets to aid the reader, or again put sub-headings in the parameter tables.

  *Paragraph was removed during revisions*

- As you did not have a strong difference in predictive capability between experiments with and without nutrient or water addition, even though you had different parameters, that presumably means you have a certain amount of model equifinality? You discuss and show the difference in model behavior as a result of the different approaches in Figures 5 – 7, but you do not discuss which one you think leads to the right behav- ior? Do you have an idea? Perhaps a synthetic experiment with pseudo-observations taken from the model simulations might help with this (a so-called "observing system simulation experiment", or OSSE)?

*This was a very insightful comment. Our response reflects the updated analysis described above that has two assimilation approaches: with and without ecosystem experiments. Our new Figure 5 (the bar graph with the experimental responses from the observations and model predictions) helps support the following:*
- *Including experiments in the assimiliation substantially increases the predictive capacity of the model in the $CO_2$ experiments.*
- *The predictive capacity of drought, irrigation, and nutrient fertilization experiments did not substantially change whether experiments where included or not.*

*We think that an OSSE would be a great follow on study that more specifically explores of the issues that are brought up in this analysis. An OSSE could explore how locations of plots within a region and the different types of individual experiments influence the ability to retrieve known parameters. Such a study would build on the description of the cost function and general approach presented in this manuscript. Since we do not include an OSSE, we now try to avoid making general statements in the discussion that would require an OSSE to quantitatively support.*

- Lines 522 onwards show very interesting results. However, I would suggest that the patterns detailed in last two paragraphs (Lines 553-572) would benefit from explanations linking back a bit more (not just referring to figures) to the different model behavior/mechanisms identified and discussed in the RW-fert and RW-water sections just above.

  *In response to Reviewer 1, we cut the results of the regional simulations from the RW-fert and RW-water simulations*

Discussion

- First paragraph is more of a summary than a discussion and could be cut or added to conclusions.

  *We prefer to provide a summary at the beginning of discussions to remind the reviewer of key points.*

- Although perhaps a little too long, this is a useful discussion that ties the results to- gether and answers some of the questions I raised in my comments on the results. Perhaps it would be useful to combine some of the summary points raised in the dis- cussion with relevant sections in the results with separate sub-headings as I mentioned above.

  *We added subheadings to the discussion*

- Lines 650-652: Interesting point and in addition, as I have mentioned above, I think a synthetic experiment would also be very helpful in this regard.

  *We agree that a synthetic experiment would be an excellent next study. The synthetic experiment could create 'fake' region with different environmental gradients and explore the types of gradients that allow for the retrieval of parameters from the OSSE study.*

Minor comments

- Line 87: Do you mean the "assimilation of manipulation experimental data", rather than the "assimilation of experiments"?

  *Yes. Fixed*

- Line 88: two or more

  *Fixed*

[revised manuscript text omitted]

**Formatted Table**

Formatted Table ... [39]
Deleted Cells ... [40]
Formatted Table ... [41]
Deleted Cells ... [47]
Formatted Table ... [48]
Moved (insertion) [13] ... [49]
Inserted Cells ... [50]
Deleted Cells ... [55]
Deleted Cells ... [57]
Moved down [15]: 0.50
Moved down [14]: 0.51
Moved (insertion) [15] ... [58]
Moved (insertion) [14] ... [59]
Moved (insertion) [16] ... [60]
Formatted Table ... [61]
Deleted Cells ... [67]
Moved down [17]: 0.76

Table 5. The optimized medians, range of the 99% quantile intervals of the posterior distributions and the 99% quantile range for priors with normally distributed priors or the range of the upper and lower bounds for priors with uniform distributions.

| Parameter | Posterior median | Posterior 99% C.I. range | Prior range | Posterior/Prior Range |
|---|---|---|---|---|
| Allocation and structure | | | | Parameter group mean = 0.38 |
| pFS2 | 0.58 | 0.55 - 0.61 | 0.08 – 1.00 | 0.06 |
| pFS20 | 0.57 | 0.55 - 0.59 | 0.10 – 1.00 | 0.05 |
| pR | 0.11 | 0.07 - 0.15 | 0.05 – 2.00 | 0.04 |
| pCRS | 0.26 | 0.25 - 0.27 | 0.15 - 0.35 | 0.11 |
| pCRS (Duke) | 0.21 | 0.18 - 0.23 | 0.15 - 0.35 | 0.20 |
| SLA0 | 8.44 | 7.67 - 9.25 | 4.4 - 6.66 | 0.70 |
| SLA1 | 2.84 | 2.72 - 2.96 | 3.59 - 4.16 | 0.43 |
| tSLA | 4.13 | 3.88 - 4.41 | 0.43 - 11.51 | 0.05 |
| fCpFS700 | 0.74 | 0.60 - 0.90 | 0.50 – 1.00 | 0.60 |
| StemConst | 0.022 | 0.009 - 0.035 | 0.009 - 0.035 | 1.00 |
| StemPower | 2.78 | 2.29 - 3.27 | 2.25 - 3.29 | 0.95 |
| Canopy photosynthesis, autotrophic respiration, and transpiration | | | | Parameter group mean = 0.14 |
| alpha | 0.029 | 0.026 - 0.031 | 0.02 - 0.06 | 0.14 |
| y | 0.50 | 0.47 - 0.53 | 0.30 - 0.65 | 0.15 |
| MaxCond | 0.011 | 0.01 - 0.012 | 0.005 - 0.03 | 0.09 |
| LAIgcx | 2.2 | 2.0 - 2.48 | 2.0 - 5.0 | 0.16 |
| Environmental modifiers of photosynthesis and transpiration | | | | Parameter group mean = 0.61 |
| kF | 0.16 | 0.12 - 0.2 | 0.14 - 0.22 | 1.04 |
| Tmin | -5.56 | -8.88 - -2.69 | -1.15 - 9.15 | 0.60 |
| Topt | 23.42 | 21.1 - 26.31 | 19.85 - 30.15 | 0.51 |
| Tmax | 39.56 | 34.71 - 44.39 | 32.85 - 43.15 | 0.94 |
| SWconst | 1.09 | 0.91 - 1.56 | 0.01 - 1.8 | 0.36 |
| SWpower | 8.86 | 3.39 - 12.98 | 1.00 – 13.00 | 0.80 |
| CoeffCond | 0.036 | 0.029 - 0.043 | 0.034 - 0.048 | 0.91 |
| fCalpha700 | 1.33 | 1.18 - 1.52 | 1.0 - 1.80 | 0.43 |
| MaxAge | 151.5 | 54.4 - 199.6 | 16.0 - 200.0 | 0.79 |

| | | | | |
|---|---|---|---|---|
| nAge | 3.35 | 1.77 - 3.99 | 1.00 – 4.00 | 0.74 |
| rAge | 2.25 | 0.81 - 2.99 | 0.01 – 3.00 | 0.73 |
| FR1 | 0.073 | 0.061 - 0.086 | 0.00 – 1.00 | 0.03 |
| FR2 | 0.17 | 0.15 - 0.19 | 0.0 – 1.0 | 0.04 |
| Mortality | | | Parameter group mean = 0.37 | |
| wSx1000 | 176.9 | 169.6 - 184.4 | 165.6 - 294.4 | 0.15 |
| wSx1000 (Duke) | 243.3 | 196.89 - 305.02 | 165.6 - 294.4 | 0.76 |
| ThinPower | 1.68 | 1.60 - 1.78 | 1.00 - 2.5 | 0.12 |
| ThinPower v(Duke) | 1.26 | 1.00 - 1.85 | 1.00 - 2.5 | 0.56 |
| mS | 0.52 | 0.37 - 0.71 | 0.10 – 1.00 | 0.38 |
| Rttover | 0.023 | 0.017 - 0.031 | 0.017 - 0.042 | 0.55 |
| MortRate | 0.001 | 9e-04 - 0.0011 | 2e-04 - 0.004 | 0.06 |
| Understory hardwoods | | | Parameter group mean = 0.28 | |
| alpha_h | 0.02 | 0.02 - 0.02 | 0.005 - 0.07 | 0.01 |
| pFS_h | 1.78 | 1.54 - 2.06 | 0.2 – 3.0 | 0.19 |
| pR_h | 0.21 | 0.06 - 0.43 | 0.05 – 2.00 | 0.19 |
| SLA_h | 16.3 | 14.1 – 19.0 | 6.2 - 25.8 | 0.25 |
| fCalpha700_h | 1.84 | 1.58 - 2.17 | 1.0 – 2.50 | 0.74 |

Inserted Cells ... [73]
Deleted Cells ... [79]
Moved up [16]: 1.09
Deleted Cells ... [82]
Inserted Cells ... [86]
Deleted Cells ... [90]
Deleted Cells ... [92]
Deleted Cells ... [103]
Formatted Table ... [97]
Moved (insertion) [17] ... [104]
Formatted Table ... [105]
Deleted Cells ... [111]
Moved (insertion) [18] ... [112]
Formatted Table ... [113]
Moved up [13]: 0.26
Inserted Cells ... [115]
... [117]
... [118]
Deleted Cells ... [119]
... [120]
Deleted Cells ... [121]
... [122]

[revised manuscript text omitted]

Formatted Table

| Page 44: [40] Deleted Cells | | Revisions | | 5/22/17 1:33:00 PM |
|---|---|---|---|---|

Deleted Cells

| Page 44: [41] Formatted Table | | Revisions | | 5/22/17 1:33:00 PM |
|---|---|---|---|---|

Formatted Table

| Page 44: [42] Deleted | | Revisions | | 5/22/17 1:33:00 PM |
|---|---|---|---|---|

(0.034 – 0.040)

| Page 44: [43] Deleted | | Revisions | | 5/22/17 1:33:00 PM |
|---|---|---|---|---|

(

| Page 44: [43] Deleted | | Revisions | | 5/22/17 1:33:00 PM |
|---|---|---|---|---|

(

| Page 44: [44] Deleted | Revisions | 5/22/17 1:33:00 PM |

(0.035 – 0.040)

| Page 44: [45] Deleted | Revisions | 5/22/17 1:33:00 PM |

(0.030 – 0.042)

| Page 44: [46] Deleted | Revisions | 5/22/17 1:33:00 PM |

0.032
(0.030 – 0.035)

| Page 44: [47] Deleted Cells | Revisions | 5/22/17 1:33:00 PM |

Deleted Cells

| Page 44: [48] Formatted Table | Revisions | 5/22/17 1:33:00 PM |

Formatted Table

| Page 44: [49] Moved from page 0 (Move #13) Revisions | | 5/22/17 1:33:00 PM |

0.26

| Page 44: [50] Inserted Cells | Revisions | 5/22/17 1:33:00 PM |

Inserted Cells

| Page 44: [51] Deleted | Revisions | 5/22/17 1:33:00 PM |

(

| Page 44: [51] Deleted | Revisions | 5/22/17 1:33:00 PM |

(

| Page 44: [52] Deleted | Revisions | 5/22/17 1:33:00 PM |

(

| Page 44: [52] Deleted | Revisions | 5/22/17 1:33:00 PM |

(

| Page 44: [53] Deleted | Revisions | 5/22/17 1:33:00 PM |

(0.46 – 0.51)

| Page 44: [54] Deleted | Revisions | 5/22/17 1:33:00 PM |

0.48
(0.45 – 0.51)

| Page 44: [55] Deleted Cells | Revisions | 5/22/17 1:33:00 PM |

Deleted Cells

| Page 44: [56] Deleted | Revisions | 5/22/17 1:33:00 PM |

0.52
(

| Page 44: [57] Deleted Cells | Revisions | 5/22/17 1:33:00 PM |

Deleted Cells

| Page 44: [58] Moved from page 44 (Move #15)Revisions | 5/22/17 1:33:00 PM |
|---|---|

0.50

| Page 44: [59] Moved from page 44 (Move #14)Revisions | 5/22/17 1:33:00 PM |
|---|---|

0.51

| Page 44: [60] Moved from page 0 (Move #16) Revisions | 5/22/17 1:33:00 PM |
|---|---|

1.09

| Page 44: [61] Formatted Table | Revisions | 5/22/17 1:33:00 PM |
|---|---|---|

Formatted Table

| Page 44: [62] Deleted | Revisions | 5/22/17 1:33:00 PM |
|---|---|---|

(1.22 – 1.40)

| Page 44: [63] Deleted | Revisions | 5/22/17 1:33:00 PM |
|---|---|---|

(

| Page 44: [63] Deleted | Revisions | 5/22/17 1:33:00 PM |
|---|---|---|

(

| Page 44: [64] Deleted | Revisions | 5/22/17 1:33:00 PM |
|---|---|---|

(

| Page 44: [64] Deleted | Revisions | 5/22/17 1:33:00 PM |
|---|---|---|

(

| Page 44: [65] Deleted | Revisions | 5/22/17 1:33:00 PM |
|---|---|---|

1.32
(1.23 – 1.41)

| Page 44: [66] Deleted | Revisions | 5/22/17 1:33:00 PM |
|---|---|---|

1.11
(1.08 – 1.15)

| Page 44: [67] Deleted Cells | Revisions | 5/22/17 1:33:00 PM |
|---|---|---|

Deleted Cells

| Page 44: [68] Deleted | Revisions | 5/22/17 1:33:00 PM |
|---|---|---|

0.84
(0.75 – 0.93)

| Page 44: [69] Deleted | Revisions | 5/22/17 1:33:00 PM |
|---|---|---|

0.83
(0.75 – 0.93)

| Page 44: [70] Deleted | Revisions | 5/22/17 1:33:00 PM |
|---|---|---|

.84
(

| Page 44: [70] Deleted | Revisions | 5/22/17 1:33:00 PM |
|---|---|---|

.84
(

| Page 44: [71] Deleted | Revisions | 5/22/17 1:33:00 PM |
|---|---|---|

(

| Page 44: [72] Deleted | Revisions | 5/22/17 1:33:00 PM |
|---|---|---|

0.99
(0.95 – 1.0)

| Page 45: [73] Inserted Cells | Revisions | 5/22/17 1:33:00 PM |
|---|---|---|

Inserted Cells

| Page 45: [74] Deleted | Revisions | 5/22/17 1:33:00 PM |
|---|---|---|

(1.09 – 1.85)

| Page 45: [75] Deleted | Revisions | 5/22/17 1:33:00 PM |
|---|---|---|

(0.95 – 1.70)

| Page 45: [76] Deleted | Revisions | 5/22/17 1:33:00 PM |
|---|---|---|

1.8
(1.47 – 2.15)

| Page 45: [77] Deleted | Revisions | 5/22/17 1:33:00 PM |
|---|---|---|

1.30
(0.89 – 1.76)

| Page 45: [78] Deleted | Revisions | 5/22/17 1:33:00 PM |
|---|---|---|

1.57
(1.08 – 1.79)

| Page 45: [79] Deleted Cells | Revisions | 5/22/17 1:33:00 PM |
|---|---|---|

Deleted Cells

| Page 45: [80] Deleted | Revisions | 5/22/17 1:33:00 PM |
|---|---|---|

1.61
(0.90 –

| Page 45: [80] Deleted | Revisions | 5/22/17 1:33:00 PM |
|---|---|---|

1.61
(0.90 –

| Page 45: [81] Deleted | Revisions | 5/22/17 1:33:00 PM |
|---|---|---|

1.29
(0.78 – 1.98)

| Page 45: [82] Deleted Cells | Revisions | 5/22/17 1:33:00 PM |
|---|---|---|

Deleted Cells

| Page 45: [83] Deleted | Revisions | 5/22/17 1:33:00 PM |
|---|---|---|

(1.48 – 3.82)

| Page 45: [84] Deleted | Revisions | 5/22/17 1:33:00 PM |
|---|---|---|

2.20
(1.47

| Page 45: [84] Deleted | Revisions | 5/22/17 1:33:00 PM |
|---|---|---|

2.20
(1.47

| Page 45: [85] Deleted | Revisions | 5/22/17 1:33:00 PM |
|---|---|---|

1.47
(

| Page 45: [86] Inserted Cells | Revisions | 5/22/17 1:33:00 PM |
|---|---|---|

Inserted Cells

| Page 45: [87] Deleted | Revisions | 5/22/17 1:33:00 PM |
|---|---|---|

(

| Page 45: [87] Deleted | Revisions | 5/22/17 1:33:00 PM |
|---|---|---|

(

| Page 45: [88] Deleted | Revisions | 5/22/17 1:33:00 PM |
|---|---|---|

(0.088 – 0.103)

| Page 45: [89] Deleted | Revisions | 5/22/17 1:33:00 PM |
|---|---|---|

(0.110 – 0.128)

| Page 45: [90] Deleted Cells | Revisions | 5/22/17 1:33:00 PM |
|---|---|---|

Deleted Cells

| Page 45: [91] Deleted | Revisions | 5/22/17 1:33:00 PM |
|---|---|---|

0.094
(0.087 – 0.102)

| Page 45: [92] Deleted Cells | Revisions | 5/22/17 1:33:00 PM |
|---|---|---|

Deleted Cells

| Page 45: [93] Deleted | Revisions | 5/22/17 1:33:00 PM |
|---|---|---|

(0.133 – 0.154)

| Page 45: [94] Deleted | Revisions | 5/22/17 1:33:00 PM |
|---|---|---|

(

| Page 45: [94] Deleted | Revisions | 5/22/17 1:33:00 PM |
|---|---|---|

(

| Page 45: [95] Deleted | Revisions | 5/22/17 1:33:00 PM |
|---|---|---|

(

| Page 45: [95] Deleted | Revisions | 5/22/17 1:33:00 PM |
|---|---|---|

(

| Page 45: [95] Deleted | Revisions | 5/22/17 1:33:00 PM |
|---|---|---|

(

| Page 45: [96] Deleted | Revisions | 5/22/17 1:33:00 PM |
|---|---|---|

0.153
(0.140 – 0.168)

| Page 45: [97] Formatted Table | Revisions | 5/22/17 1:33:00 PM |
|---|---|---|

Formatted Table

| Page 45: [98] Deleted | Revisions | 5/22/17 1:33:00 PM |
|---|---|---|

(171 – 181)

| Page 45: [99] Deleted | Revisions | 5/22/17 1:33:00 PM |
|---|---|---|

(174 – 186)

| Page 45: [100] Deleted | Revisions | 5/22/17 1:33:00 PM |
|---|---|---|

(176 – 186)

| Page 45: [101] Deleted | Revisions | 5/22/17 1:33:00 PM |
|---|---|---|

(228 – 295)

| Page 45: [102] Deleted | Revisions | 5/22/17 1:33:00 PM |
|---|---|---|

(174 0 187)

| Page 45: [103] Deleted Cells | Revisions | 5/22/17 1:33:00 PM |
|---|---|---|

Deleted Cells

| Page 45: [104] Moved from page 44 (Move #17)Revisions | | 5/22/17 1:33:00 PM |
|---|---|---|

0.76

| Page 45: [105] Formatted Table | Revisions | 5/22/17 1:33:00 PM |
|---|---|---|

Formatted Table

| Page 45: [106] Deleted | Revisions | 5/22/17 1:33:00 PM |
|---|---|---|

(1.60 – 1.74)

| Page 45: [107] Deleted | Revisions | 5/22/17 1:33:00 PM |
|---|---|---|

(1.63 –

| Page 45: [107] Deleted | Revisions | 5/22/17 1:33:00 PM |
|---|---|---|

(1.63 –

| Page 45: [108] Deleted | Revisions | 5/22/17 1:33:00 PM |
|---|---|---|

(1.65 – 1.78)

| Page 45: [109] Deleted | Revisions | 5/22/17 1:33:00 PM |
|---|---|---|

1.28
(1

| Page 45: [109] Deleted | Revisions | 5/22/17 1:33:00 PM |
|---|---|---|

1.28

(1

| Page 45: [110] Deleted | Revisions | 5/22/17 1:33:00 PM |

1.61
(1.51 – 1.69)

| Page 45: [111] Deleted Cells | Revisions | 5/22/17 1:33:00 PM |

Deleted Cells

| Page 45: [112] Moved from page 45 (Move #18)Revisions | 5/22/17 1:33:00 PM |

0.19

| Page 45: [113] Formatted Table | Revisions | 5/22/17 1:33:00 PM |

Formatted Table

| Page 45: [114] Deleted | Revisions | 5/22/17 1:33:00 PM |

(0.25 – 0.27)

| Page 45: [115] Inserted Cells | Revisions | 5/22/17 1:33:00 PM |

Inserted Cells

| Page 45: [116] Deleted | Revisions | 5/22/17 1:33:00 PM |

0.24
(0.23 – 0.

| Page 45: [116] Deleted | Revisions | 5/22/17 1:33:00 PM |

0.24
(0.23 – 0.

| Page 45: [117] Deleted | Revisions | 5/22/17 1:33:00 PM |

(0.24 – 0.26)

| Page 45: [118] Deleted | Revisions | 5/22/17 1:33:00 PM |

0.17
(0.16 –

| Page 45: [119] Deleted Cells | Revisions | 5/22/17 1:33:00 PM |

Deleted Cells

| Page 45: [120] Deleted | Revisions | 5/22/17 1:33:00 PM |

0.28
(0.27 – 0.29)

| Page 45: [121] Deleted Cells | Revisions | 5/22/17 1:33:00 PM |

Deleted Cells

| Page 45: [122] Deleted | Revisions | 5/22/17 1:33:00 PM |

Figures

[Figure]

Figure 1.

[Figure]

[Figure]

Figure 3.

change in stem biomass of a 25-year stand when nutrient limitation is completely removed through nutrient addition (simulated by setting FR = 1). Predictions from data assimilation that included nutrient addition experiments are shown in (a) and prediction data assimilation that did not include nutrient addition experiments are shown in (b). The focal site in Georgia highlighted in Figures 5c and 6b is represented by the circle containing the dot.

[Figure]

Figure 11. Regional predictions of the change in stem biomass of a 25-year stand when annual precipitation is reduced by 30%. Predictions from data assimilation that included water manipulation experiments are shown in (a) and prediction data assimilation that did not include water manipulation experiments are shown in (b). The focal site in Georgia highlighted in Figures 6c and 7b is represented by the circle containing the dot.